# Synthesis, Antitumor Evaluation, Molecular Modeling and Quantitative Structure–Activity Relationship (QSAR) of Novel 2-[(4-Amino-6-*N*-substituted-1,3,5-triazin-2-yl)methylthio]-4-chloro-5-methyl-*N*-(1*H*-benzo[*d*]imidazol-2(3*H*)-ylidene)Benzenesulfonamides

**DOI:** 10.3390/ijms21082924

**Published:** 2020-04-22

**Authors:** Łukasz Tomorowicz, Jarosław Sławiński, Beata Żołnowska, Krzysztof Szafrański, Anna Kawiak

**Affiliations:** 1Department of Organic Chemistry, Medical University of Gdańsk, Al. Gen. J. Hallera 107, 80-416 Gdansk, Poland; ltomorowicz@gumed.edu.pl (Ł.T.); k.szafranski@gumed.edu.pl (K.S.); 2Department of Biotechnology, Intercollegiate Faculty of Biotechnology, University of Gdańsk and Medical University of Gdańsk, ul. Abrahama 58, 80-307 Gdansk, Poland; anna.kawiak@biotech.ug.edu.pl

**Keywords:** benzenesulfonamide, synthesis, 1,3,5-triazines, cytotoxicity, QSAR, molecular docking

## Abstract

A series of novel 2-[(4-amino-6-R^2^-1,3,5-triazin-2-yl)methylthio]-4-chloro-5-methyl-*N*-(5-R^1^-*1H*-benzo[*d*]imidazol-2(*3H*)-ylidene)benzenesulfonamides **6**–**49** was synthesized by the reaction of 5-substituted ethyl 2-{5-R^1^-2-[*N*-(5-chloro-*1H*-benzo[*d*]imidazol-2(*3H*)-ylidene)sulfamoyl]-4-methylphenylthio}acetate with appropriate biguanide hydrochlorides. The most active compounds, **22** and **46**, showed significant cytotoxic activity and selectivity against colon (HCT-116), breast (MCF-7) and cervical cancer (HeLa) cell lines (IC_50_: 7–11 µM; 15–24 µM and 11–18 µM), respectively. Further QSAR (Quantitative Structure–Activity Relationships) studies on the cytotoxic activity of investigated compounds toward HCT-116, MCF-7 and HeLa were performed by using different topological (2D) and conformational (3D) molecular descriptors based on the stepwise multiple linear regression technique (MLR). The QSAR studies allowed us to make three statistically significant and predictive models for them. Moreover, the molecular docking studies were carried out to evaluate the possible binding mode of the most active compounds, **22** and **46**, within the active site of the MDM2 protein.

## 1. Introduction

Cancer is a major public health problem worldwide and is the second leading cause of death in developed nations. The greatest number of deaths are from cancers of the lung, prostate, colon and rectum in men and the lung, breast, colon and rectum in women [1]. One of the basic methods of cancer treatment is chemotherapy, which uses cytotoxic drugs with systemic effects. Despite years of effort in the field of designing different molecules, there are still few selective drugs against cancer cells as compared with normal cells [2].

The mouse/murine protein, MDM2, a promising target for developing anti-cancer therapies, is an important negative regulator of the p53 tumor suppressor protein [3,4]. Under normal conditions, the MDM2 protein binds to the transactivation domain of p53, preventing its binding to DNA and labelling DNA for proteasomal degradation. In this way, MDM2/p53 interaction reduces p53 abundance in normal and untransformed cells [5,6]. Conversely, in several cancer cells, MDM2 has been shown to be overexpressed and leads to a loss of the tumor-suppressor function of p53, promoting proliferation, survival and growth of the tumor [7,8,9,10]. Preclinical data have shown that blocking the MDM2/p53 interaction may induce apoptosis in both MDM2-overexpressing and wild-type tumor cell lines [11]. Hence, small molecules designed to block the MDM2/p53 interaction can lead to an increase in the level of p53 and its transcriptional activation [3].

The nutlins are the first class of potent and specific MDM2 small-molecule inhibitors, published by Vassilev in 2004. They are analogs of cis-imidazoline and are capable of binding to MDM2 in the p53-binding pocket, activating the p53 pathway in cancer cells [12]. The most extensively investigated molecule belonging to cis-imidazoline derivatives is Nutlin-3a, the most active and potent enantiomer of the nutlins family (Figure 1). Preclinical evaluations have widely demonstrated, in both in vitro and in vivo tumor models, that Nutlin-3a showed antitumor activity against breast cancer, melanoma, retinoblastoma, prostate cancer, lymphoma, and hematological malignancies [13]. The nutlins as promising compounds, which were entered into clinical trials, became interesting lead structures for further chemical modifications.

Our previous works on a search for antitumor agents among benzenesulfonamide derivatives, carried out by Sławinski’s group, indicate the importance of the 2-methylthiobenzenesulfonamide fragment for cytotoxic activity of compounds against cervical, breast and colon cancer. We have proved that our compounds showed an apoptotic effect in cancer cells. Continuing the search for more active compounds, we designed and developed a method for the synthesis of new molecules with potential inhibitory activity against the MDM2 protein. We carried out molecular docking for various targets associated with tumors that showed the affinity of designed compounds for MDM2. In this work, we report on a series of 2-[(4-amino-6-R^2^-1,3,5-triazin-2-yl)methylthio]- -4-chloro-5-methyl-*N*-(5-R^1^-1*H*-benzo[*d*]imidazol-2(3*H*)-ylidene)benzenesulfonamides designed as molecular hybrids combining the 2-mercaptobenzenosulfonamide fragment with the imidazoline ring (Figure 1). In the structure of our compounds, we also incorporated different substituents R^2^ to investigate their impact on anticancer activity and establish structure-activity relationships. All compounds were tested for their cytotoxic activity against HCT-116, MCF-7 and HeLa cell lines.

## 2. Results and Discussions

### 2.1. Chemistry

The starting substrates 3-amino-6-chloro-7-methyl-1,4,2-benzodithiazine (**1**), ethyl 2-[5-chloro-2-(*N*-cyanosulfamoyl)-4-methylphenylthio]acetate potassium salt (**2**) [14], and most of the biguanide hydrochlorides, were prepared according to known methods [15,16,17,18,19,20]. Novel substrates **3**–**5** were synthesized analogously by the reaction of **2** with 4-R^1^-benzene-1,2-diamine as shown in Scheme 1. Finally 2-[(4-amino-6-R^2^-1,3,5-triazin-2-yl)methylthio]-4-chloro-5-methyl- -*N*-(5-R^1^-*1H*-benzo[*d*]imidazol-2(*3H*)-ylidene)benzenesulfonamides (**6**–**49**) were obtained by the reaction of esters **3**–**5** with appropriate biguanide hydrochlorides in MeONa/MeOH solution at reflux for 45 h.

The final compounds were characterized by IR and NMR (see representative spectra in Appendix A) spectroscopy as shown in the Experimental Section. Elemental analyses (C, H, N) and HRMS were in accordance with the proposed structures.

For example, in the ^1^H NMR spectra the presence of a S-CH_2_-1,3,5-triazine ring and NH groups in the benzimidazole rings of **6**–**49** were identified from the singlet signals at 3.81–4.05 and two signals at 10.5–12.15 ppm, respectively. Meanwhile the appearance of H-3 and H-6 of the benzene ring at 7.61–7.94 and 7.90–8.00 ppm, respectively, confirmed the proposed structure of final compounds **6**–**49**.

### 2.2. Cytotoxic Activity

Compounds **6**–**49** were evaluated in vitro for their effects on the viability of the three human cancer cell lines: HCT-116 (colon cancer), MCF-7 (breast cancer) and HeLa (cervical cancer) as well as the non-cancerous keratinocyte cell line (HaCaT). The concentration required for 50% inhibition of cell viability IC_50_ was calculated and compared with the reference drug cisplatin, the results are shown in Table 1.

Cell lines: colon cancer (HCT-116), breast cancer (MCF-7), cervical cancer (HeLa), the human keratinocyte cell line (HaCaT); NT—not tested; IC_50_ was measured at concentrations 1, 10, 25, 50, and 100 µM. IC_50_ values are expressed as the mean ± SD of at last three independent experiments.

The most active compound **46** belonged to the 5-chlorobenzimidazole series (R^2^ = [4-(4-fluorophenyl)piperazin-1-yl] and showed outstanding activity (7 µM, 15 µM and 18 µM, respectively) against all tested cell lines HCT-116, MCF-7 and HeLa with selectivity ratios HaCaT/HCT-116 and HaCaT/MCF-7 in the range of 4 to 2 (Figure 2). Moreover, compound **22** belonged to the benzimidazole series (*R*^2^ = [4-(4-trifluoromethylphenyl)piperazin-1-yl] and strongly inhibited HCT-116 and HeLa cell line viability (IC_50_ = 11 µM) with selectivity ratios HaCaT/HCT-116 or HeLa equal to 3.1 (Figure 2).

As shown in Table 1, the HCT-116 cell line presented the relatively highest susceptibility and was affected by eighteen compounds **19**–**26**, **28**, **30**, **31**, **34**, **37**, **38**, **40**, **42**, **45** and **46,** in the range of IC_50_ values from 7 to 18 µM. Meanwhile, the HeLa cell line was susceptible towards nine compounds (**19**–**23, 30, 38**, **40** and **46**) with IC_50_ values of 11–18 µM, and the MCF-7 by four compounds (**19**, **20**, **23** and **46**) with IC_50_ values ranging between 17 and 19 µM.

We found that among the series bearing a 4-arylpiperazine moiety, the presence of *R*^2^ = 4-Ph-piperazin-1-yl (**19**, **46**), 4-(4-flurophenylpiperazin-1-yl (**20**), and 4-(4-chlorophenylpiperazin-1-yl (**23**) substituents provided strong cytotoxicity toward all of the tested cell lines with IC_50_ values in the range of 7–19 µM, while replacement of an aryl group by a methyl **18** (*R*^2^ = 4-methylpiperazin-1-yl) or phenylsulfonyl moiety **32** (*R*^2^ = 4-(phenylsulfonyl)piperazin-1-yl) caused a decrease in activity to IC_50_ values of 42–82 µM (**18**) or the loss of activity (IC_50_ 137–280 µM) for **32** (Table 1). It should be mentioned that for the series without the piperazine moiety at position 6 of the 1,3,5-triazine ring **6**–**17, 33**–**36** and **41**–**44**, the presence of the indoline moiety **9**, **34** (*R*^2^ = indolin-1-yl) or anilino group **42** (*R*^2^ = PhNH-) resulted in moderate cytotoxic activity toward all tested cell lines (IC_50_ 17–30 µM), at the same time showing an increase in activity towards HCT-116 (IC_50_ 17–18 µM) for **34** and **42**, respectively. On the other hand, replacement of anilino moiety **11** (*R*^2^ = PhNH-) by benzylamino group **15** (*R*^2^ = PhCH_2_NH-) caused a 1.7-fold increase in activity toward HCT-116, as well as a 1.5-fold increase toward MCF-7 cells (Table 1).

### 2.3. Quantitative Structure–Activity Relationships (QSARs) of Cytotoxic Activity

Correlation between structure and activity was performed according to QSAR methodology. Three-dimensional structures of all compounds were obtained by applying a conformational search with the LowModeMD method (MOE software) using MMFF94X forcefield (MOE software) followed by geometry optimization with the semi empirical PM6 method (MOPAC2016 software). The energy of final structures were calculated using GAMESS software and the STO3G HF method. Molecular descriptors were calculated using MOE software. In order to obtain QSAR models, stepwise linear progressive regression (a type of MLR) was applied. Compounds with activity above 100 µM were removed from the model development (**29** and **32** for the HCT-116 cell line, **7**, **18**, **27**, **29**, **32**, **39** for the MCF-7 cell line and **7**, **27**, **29**, **15**, **32** for the HeLa cell line).

The obtained models correlated cytotoxic activity (IC_50_) toward cancer cells with different topological (2D) and conformational (3D) molecular descriptors. The equations were statistically significant, they explained 75–86% of the variability of the IC_50_ coefficient and were characterized by usefulness of model to predict the antitumor activity of new sulfonamides, as indicated by the values of Q^2^ from 68% to 75% (Table 2 and Figure 3).

For each model, a residue analysis was carried out to confirm the correctness of used linear regression and to confirm its assumptions (such as demonstration of an absence of deviations from linearity, and normality of residue distribution to confirm homoscedasticity). The predictors that corresponded most with antitumor activity were estimated: a_nO (number of oxygen atoms in the molecule) for the HCT-116 model with a correlation coefficient of 0.50, SMR_VSA0 (adjacency and distance matrix descriptor) for the MCF-7 model with a correlation coefficient of 0.75 and for the HeLa model there is SlogP_VSA5, which represents different aspects of the van der Waals surface area’s contribution to lipophilicity with correlation coefficient of 0.55.

From the HCT-116 model, it is clear that higher activity correlates with lower values of the number of oxygen atoms (**a_nO**), third diagonal element of diagonalized moment of inertia tensor (pmi3), out-of-plane potential energy (E_oop), length of the longest single bond chain (b_max1len), hydrophilic integy moment (**vsurf_IW6**). On the other hand, the negative coefficient of GCUT_SLOGP_1 shows that the high value of this descriptor is valuable for anticancer activity. The cytotoxic activity of the compounds against MCF-7 has correlation with six descriptors. Two beneficial impacts were shown: atom information content (**a_IC**) and shape (**std_dim3**) descriptors, which prefer high values and **b_max1len**, **GCUT_SLOGP_2**, **PEOE_VSA+1, SMR_VSA0** descriptors favoring low values. In the HeLa model, it can be noticed that the increase of biological activity relates to higher values of both parameters: the number of fluorine atoms (**a_nF**) and number of rotatable single bonds (**b_1rotN**). Increased values of descriptors related to atom counts and bond counts (**ast_violation**), Huckel theory (**h_pstrain**), and the structure connectivity and conformation (**pmi, SlogP_VSA5**) decrease the anticancer activity of molecules.

### 2.4. Molecular Modeling and Docking Results

In order to better understanding the anticancer activity of synthesized compounds, molecular docking was carried out for various therapeutic targets of cancer. It was found that the proper fitting with good energy scores was shown for the MDM2 protein, while the majority of the compounds had a moderate score with other targets, e.g., serine-threonine protein kinases Akt-1 [21], RAF [22] and B-RAF [22] or epidermal growth factor receptor EGFR [23] among others.

Molecular docking of some of the newly synthesized compounds within the active site of the MDM2 protein was performed and the amino acid interactions and docking patterns were investigated using the protein data bank file (PDB ID:5C5A). This file contains the MDM2 protein co-crystalized with Nutlin-3a. The docking procedures were performed by Molecular Operating Environment (MOE, 2018) software. The docking setup was first validated by self-docking of the co-crystallized ligand (Nutlin-3a) in the binding site of the protein, with energy score S = −10.8029 kcal/mol and root mean standard deviation (RMSD) = 0.2534. The ligand interacts with Met62, His96, Gly58, Gln59, Leu54 and Val93 in the active site of MDM2 (Figure 4).

Docking of the most active compounds **22**, **46** was performed and showed proper fitting in the active site of MDM2 with positive energy scores (S), which supports the observed activity of these compounds as MDM2 inhibitors. The energy score (S) and amino acid interaction of the most potent MDM2 inhibitors are listed in Table 3. The docking results revealed that the amino acids Leu54 and Met62 located in the binding pocket of the protein played an important role. Thus, the most active compounds (**22**, **46**) showed interaction with Leu54 and Met62 formed π-H interaction with Leu54 and/or H-bond donor with Met62, which mimics the pattern of interaction of Nutlin-3a with the MDM2 protein (Figure 4). Interestingly, the 2-(4-phenylpiperazin-1-yl)-1,3,5-triazine fragment is located in a hydrophobic binding pocket near amino acid residues Met62, Leu54, Val93, Gly58, and Gln59, interacting directly with the most important residues Leu54 and Met62 (Figure 5). Apparently, compounds **22** and **46** showed the same orientation within the active site of MDM2, suggesting the binding pattern of these derivatives within the MDM2 protein (Figure 6). The remaining parts of the molecules, i.e., the benzenesulfonamide fragment and the benzimidazole ring occupy other regions of the protein by interacting with the amino acids Lys51 or Met50, as well Tyr100, respectively. Based on the characterization of the protein–ligand interactions, the 4-phenylpiperazin-1-yl moiety played a key role in forming a H-bond interaction, while both 1,3,5-triazine and benzimidazole rings were responsible for π-H (polar) and aromatic π-π stacking noncovalent interactions (Table 3).

## 3. Materials and Methods

### 3.1. General Information

Melting points were measured using Stuart SMP30 (Bibby Scientific Limited, Stone Staffordshire UK) apparatus and were uncorrected. IR spectra were recorded on a Nicolet iS5 FTIR spectrometer (Thermo Fisher Scientific, Waltham, MA, USA) in KBr pellets; the absorption range was 400–4000 cm^−1^. ^1^H NMR and ^13^C NMR spectra were obtained on Varian Unity Plus 500 apparatus (Varian, Palo Alto, CA, USA). Chemical shifts are reported in parts per million (ppm). Moreover, resonance multiplicity is presented as: s (singlet), d (doublet), t (triplet), q (quartet), and m (multiplet). Elemental analyses were obtained on PerkinElmer 2400 Series II CHN Elemental Analyzer apparatus (PerkinElmer, Shelton, CT, USA) and the results indicated by the symbols of the elements were within ±0.4% of the theoretical values. Thin-layer chromatography (TLC) was conducted on Merck Kieselgel 60 F254 plates (Merck, Darmstadt, Germany) and visualized with UV. High resolution mass spectrometry (HRMS) was conducted on a TripleTOF 5600+ mass spectrometer (AB SCIEX, Framingham, MA, USA) equipped with a DuoSprayTM Ion Source and coupled with Micro HPLC system Ekspert™ microLC 200 (Eksigent Redwood City, CA, USA); Column: HALO Fused-Core C18 (50 × 0.5 mm, 2.7 μm) (Eksigent), thermostated at 50 °C; Flow: 30 μL/min; Mobile Phase: A: 0.1% formic acid in water, B: 0.1% formic acid in acetonitrile; Isocratic program 100% B, 4 min.

The following starting compounds were obtained according to the reported methods: 3-amino-6-chloro-7-methyl-1,1-dioxo-1,4,2-benzodithiazine (**1**) and ethyl 2-[5-chloro-2-(*N*-cyanosulfamoyl)-4-methylphenylthio]acetate potassium salt (**2**) [14].

### 3.2. Synthesis

#### 3.2.1. General Procedure for the Preparation of Ethyl 2-{[5-2-[*N*-(5-Chloro-1*H*-benzo[*d*]imidazol-2(3*H*)-ylidene)sulfamoyl]-4-methylphenyl]thio}acetate **3**–**5**

To the solution of ethyl 2-[5-chloro-2-(*N*-cyanosulfamoyl)-4-methylphenylthio]acetate potassium salt (**2**) (1.161 g, 3 mmol) in glacial acetic acid (25 mL) an appropriate benzene-1,2-diamine (3.15 mmol) was added. Then the reaction mixture was stirred under reflux for 7 h. After cooling, the precipitate was filtered off, and washed with glacial acetic acid (2 × 0.2 mL) and dried. The crude product was purified by crystallization from ethanol.

*Ethyl 2-{2-[N-(1H-benzo[d]imidazol-2(3H)-ylidene)sulfamoyl]-5-chloro-4-methylphenylthio}acetate* (**3**).

Starting from benzene-1,2-diamine (0.341 g, 3.15 mmol). The title compound was obtained after crystallization from ethanol. Yield 0.459 g (38%); m.p. 235–236 °C; IR (KBr): 3335 (N-H), 2980, 2892 (C-H), 1736 (C=O), 1598, 1533, 1476 (C=N, C=C_Ar_), 1294, 1138 (SO_2_), 1116 (O-CH_2_) cm^−1^; ^1^H NMR (500 MHz, DMSO-*d*_6_) δ: 1.08–1.11 (t, *J*=7.15 Hz, 3H, CH_2_-C**H_3_**), 2.33 (s, 3H, CH_3_-Ph), 3.95 (s, 2H, S-CH_2_), 4.01–4.06 (q, *J=7.1 Hz*, 2H, O-C**H**_2_CH_3_), 7.11–7.29 (m, 4H, H_Ar_), 7.42 (m, 1H, H-3), 8.02 (m, 1H, H-6), 11.96 (m, 2H, NH, benzimidazolidine) ppm; Anal.calcd. for C_18_H_18_ClN_3_O_4_S_2_ (439.94); C, 49.14; H, 4.12; N, 9.55. Found: C, 48.95; H, 3.90; N, 9.44.

*Ethyl 2-{5-fluoro-2-[N-(5-chloro-1H-benzo[d]imidazol-2(3H)-ylidene)sulfamoyl]-4-methylphenylthio} acetate* (**4**).

Starting from 4-fluorobenzene-1,2-diamine (0.397 g, 3.15 mmol) the title compound was obtained. Yield 0.414 g (30%); m.p. 219–220 °C; IR (KBr): 3334 (N-H), 2979, 2935, 2902, 2801 (C-H), 1735 (C=O), 1633, 1534, 1476 (C=N, C=C_Ar_), 1296, 1114 (SO_2_) cm^−^^1^; ^1^H NMR (500 MHz, DMSO-*d*_6_) δ: 1.08–1.11 (t, *J=7.35 Hz*, 3H, CH_2_C**H_3_**), 2.33 (m, 3H, CH_3_), 3.95 (s, 2H, S-C**H_2_**), 4.02–4.06 (m, 2H, C**H_2_**CH_3_), 6.94–7.26 (m, 3H, H_Ar_), 7.43 (m, 1H, H-3), 8.00 (m, 1H, H-6), 12.02 (m, 2H, NH, benzimidazolidine) ppm; Anal. calcd. for C_18_H_17_ClFN_3_O_4_S_2_ (457.93); C, 47.21; H, 3.74; N, 9.18. Found: C, 47.23; H, 3.68; N, 8.79.

*Ethyl 2-{5-chloro-2-[N-(5-chloro-1H-benzo[d]imidazol-2(3H)-ylidene)sulfamoyl]-4-methylphenyl-thio}acetate* (**5**).

Starting from 4-chlorobenzene-1,2-diamine (0.449 g, 3.15 mmol) the title compound was obtained. Yield 0.428 g (30%); m.p. 237–238 °C (dec.); IR (KBr): 3301 (N-H), 2983, 2928, 2857 (C-H), 1748 (C=O), 1626, 1468 (C=N, C=C_Ar_), 1281, 1146 (SO_2_), cm^−1^; ^1^H NMR (500 MHz, DMSO-*d*_6_) δ: 1.08–1.11 (t, *J*=7.1 Hz, 3H, CH_2_C**H_3_**), 2.33 (s, 3H, CH_3_), 3.95 (s, 2H, S-C**H_2_**), 4.01–4.05 (q, *J=7.1 Hz*, 2H, C**H_2_**CH_3_), 7.15–7.30 (m, 3H, H_Ar_), 7.43 (m, 1H, H-3), 7.99 (m, 1H, H-6), 12.08 (m, 2H, NH, benzimidazolidine) ppm; Anal.calcd. for C_18_H_17_Cl_2_N_3_O_4_S_2_ (474.38); C, 45.57; H, 3.61; N, 8.86. Found: C, 45.88; H, 3.61; N, 8.85.

#### 3.2.2. General Procedure for the Preparation of 6-Substituted 2-[(4-Amino-1,3,5-triazin-2-yl)methylthio]-*N*-(1*H*-benzo[*d*]imidazol-2(3*H*)-ylidene)-4-chloro-5-methylbenzenesulfonamide **6**–**32**


To the solution of sodium methoxide prepared from sodium (0.0368 g, 1.60 mmol) and anhydrous methanol (7.5 mL), ethyl 2-[{2-[*N*-(1*H*-benzo[*d*]imidazol-2(3*H*)-ylidene)sulfamoyl]-5-chloro-4-methylphenyl}thio]acetate (**3**) (0.352 g, 0.80 mmol) and the next appropriate biguanide hydrochloride (1.60 mmol) was added. The reaction mixture was stirred under reflux for 45 h. After cooling the precipitate was filtered off and dried, then stirred vigorously with water (25 mL) for 25 min. The crude product was purified by crystallization from the appropriate solvent or by extraction of the impurities with boiling ethanol, acetonitrile or diethyl ether.

*2-{[4-Amino-6-(dimethylamino)-1,3,5-triazin-2-yl]methylthio}-N-(1H-benzo[d]imidazol-2(3H)-ylidene)-4-chloro-5-methylbenzenesulfonamide* (**6**).

Starting from 1,1-dimethylbiguanide hydrochloride (0.265 g, 1.60 mmol). The title compound was obtained after extraction of the impurities with boiling ethanol (1:22). Yield 0.203 g (51%); m.p. 279–281 °C; IR (KBr): 3491, 3365, 3319 (N-H), 2949, 2925, 2887 (C-H), 1474, 1593 (C=C_Ar_), 1280, 1139 (SO_2_) cm^−^^1^; ^1^H NMR (500 MHz, DMSO-*d*_6_) δ: 2.31 (s, 3H, CH_3_Ph), 2.99 (br.s, 6H, CH_3_), 3.85 (s, 2H, S-C**H**_2_), 6.84–7.26 (m, 4H, H_Ar_ and 2H, NH_2_), 7.95 (m, 1H, H-3), 7.98 (m, 1H, H-6), 11.93 (m, 2H, NH, benzimidazolidine) ppm; ^13^C NMR (DMSO-d_6_) δ: 19.39, 36.01, 36.12, 40.21, 111.38, 122.85, 127.93, 129.88, 130.83, 131.90, 136.91, 136.98, 139.22, 150.24, 165.36, 167.12, 173.84 ppm; Anal. calcd. for C_20_H_21_ClN_8_O_2_S_2_ (505.02); C, 47.57; H, 4.19; N, 22.19. Found: C, 47.48; H, 4.13; N, 22.10.

*2-[(4-Amino-6-morpholino-1,3,5-triazin-2-yl)methylthio]-N-(1H-benzo[d]imidazol-2(3H)-ylidene)-4-chloro-5-methylbenzenesulfonamide* (**7**).

Starting from *N*-carbamimidoylmorpholine-4-carboximidamide hydrochloride (0.332 g, 1.60 mmol). The title compound was obtained after crystallization from a mixture of dimethylformamide/water (7:3). Yield 0.180 g (41%); m.p. 284–285 °C (dec.); IR (KBr): 3411, 3326, 3235 (N-H), 2969, 2905, 2855 (C-H), 1590, 1473 (C=N, C=C_Ar_), 1288, 1140 (SO_2_) cm^−^^1^; ^1^H NMR (500 MHz, DMSO-*d*_6_) δ: 2.32 (s, 3H, CH_3_), 3.54 (m, 4H, morpholine), 3.61–3.63 (m, 4H, morpholine), 3.87 (s, 2H, S-CH_2_), 6.93 (m, 2H, NH_2_ and 4H, H_Ar_), 7.92 (m, 1H, H-3), 7.99 (m, 1H, H-6), 11.93 (m, 2H, NH, benzimidazolidine) ppm; Anal. calcd. for C_22_H_23_ClN_8_O_3_S_2_ (547.05); C, 48.30; H, 4.24; N, 20.48. Found: C, 48.03; H, 4.52; N, 20.40. HRMS (ESI-TOF) 546.1023 calcd for C_22_H_23_ClN_8_O_3_S_2_ [M + H]^+^ 547.1101 found 547.1094.

*2-{[4-Amino-6-(3,5,5-trimethyl-4,5-dihydro-1H-pyrazol-1-yl)-1,3,5-triazin-2-yl]metylthio}-N-(1H-benzo[d]imidazol-2(3H)-ylidene)-4-chloro-5-methylbenzenesulfonamide* (**8**).

Starting from *N*-carbamimidoyl-3,5,5-trimethyl-4,5-dihydro-1*H*-pyrazole-1-carboximidamide hydrochloride (0.372 g, 1.60 mmol). The title compound was obtained after crystallization from ethanol (1:3), and then from acetonitrile (1:118). Yield 0.090 g (20%); m.p. 291–293 °C (dec.); IR (KBr): 3380, 3319, 3267 (N-H), 2977, 2945, 2889 (C-H), 1597, 1475 (C=N, C=C_Ar_), 1141, 1332 (SO_2_) cm^−^^1^; ^1^H NMR (500 MHz, DMSO-*d*_6_) δ: 1.22–1.47 (m, 6H, CH_3_, pyrazole), 1.93 (br.s., 2H, CH_2_, pyrazole), 2.29 (s, 3H, CH_3_Ph), 2.70 (m, 3H, CH_3_, pyrazole), 3.85–3.93 (m, 2H, S-CH_2_), 6.91–7.98 (m, 2H, NH_2_ and 4H, H_Ar_ and 1H, H-3), 8.00 (m, 1H, H-6), 11.92 (m, 2H, NH, benzimidazolidine) ppm; Anal. calcd. for C_24_H_26_ClN_9_O_2_S_2_ (572.11); C, 50.39; H, 4.58; N, 22.03. Found: C, 50.30; H, 4.55; N, 22.00. HRMS (ESI-TOF) 571.1339 calcd for C_24_H_26_ClN_9_O_2_S_2_ [M + H]^+^ 572.1417 found 572.1575.

*2-{[4-Amino-6-(indolin-1-yl)-1,3,5-triazin-2-yl]metylthio}-N-(1H-benzo[d]imidazol-2(3H)-ylidene)-4-chloro-5-methylbenzenesulfonamide* (**9**).

Starting from *N*-carbamimidoylindoline-1-carboximidamide hydrochloride (0.384 g, 1.60 mmol). The title compound was obtained after crystallization from ethanol (1:10). Yield 0.207 g (45%); m.p. 273–274 °C (dec.); IR (KBr): 3469, 3229, 3367 (N-H), 2924, 2854 (C-H), 1530, 1499 (C=N, C=C_Ar_), 1258, 1140 (SO_2_) cm^−^^1^; ^1^H NMR (500 MHz, DMSO-*d*_6_) δ: 2.26 (s, 3H, CH_3_), 3.07–3.10 (t, *J =8.6 Hz*, 2H, *3H*-indolinyl), 3.94 (m, 2H, indolinyl), 4.10 (m, 2H, S-CH_2_), 6.83–7.24 (m, 8H, H_Ar_ and 2H, NH_2_), 7.94 (m, 1H, H-3), 8.40 (m, 1H, H-6), 11.06 (m, 1H, NH, benzimidazolidine) ppm; ^13^C NMR (DMSO-d_6_) δ: 18.90, 26.44, 39.37, 47.78, 111.28, 116.70, 117.54, 119.48, 122.09, 124.64, 126.66, 126.91, 126.93, 130.76, 130.98, 132.58, 135.22, 135.86, 141.22, 142.51, 154.01, 162.66, 166.56 ppm; Anal. calcd. for C_26_H_23_ClN_8_O_2_S_2_ (579.10); C, 53.93; H, 4.00; N, 19.35. Found: C, 53.85; H, 4.06; N, 19.29.

*2-{[4-Amino-6-(3,4-dihydroquinolin-1(2H)-yl)-1,3,5-triazin-2-yl]methylthio}-N-(1H-benzo[d]imidazol-2(3H)-ylidene)-4-chloro-5-methylbenzenesulfonamide* (**10**).

Starting from *N*-carbamimidoyl-3,4-dihydroquinoline-1(*2H*)-carboximidamide hydrochloride (0.406 g, 1.60 mmol). The title compound was obtained after extraction of the impurities with boiling ethanol (1:28). Yield 0.061 g (13%); m.p. 230–232 °C; IR (KBr): 3467, 3386, 3303 (N-H), 2960, 2921, 2891, 2866 (C-H), 1562, 1509 (C=N, C=C_Ar_), 1290, 1136 (SO_2_) cm^−^^1^; ^1^H NMR (500 MHz, DMSO-*d*_6_) δ: 1.77–1.82 (pent, *J = 6.2 Hz*; 2H, H-3, dihydroquinoline), 2.32 (s, 3H, CH_3_), 2.68 (t, *J = 6.6 Hz*; 2H, H-4, dihydroquinoline), 3.85 (t, *J* = 5.8 *Hz*, 2H, H-2 dihydroquinoline), 3.93 (s, 2H, S-CH_2_), 6.92–7.67 (m, 8H, H_Ar_ and 2H, NH_2_), 7.77 (m, 1H, H-3), 8.01 (m, 1H, H-6), 11.95 (m, 2H, NH, benzimidazolidine) ppm; Anal. calcd. for C_27_H_25_ClN_8_O_2_S_2_ (593.12); C, 54.67; H, 4.25; N, 18.89. Found: C, 54.33; H, 4.00; N, 18.53.

*2-{[4-Amino-6-(phenylamino)-1,3,5-triazin-2-yl]methylthio}-N-(1H-benzo[d]imidazol-2(3H)-ylidene)-4-chloro-5-methylbenzenesulfonamide* (**11**).

Starting from 1-phenylbiguanide hydrochloride (0.342 g, 1.60 mmol). The title compound was obtained after extraction of the impurities with hot Et_2_O (1 h). Yield 0.085 g (20%); m.p. 284–285 °C (dec.); IR (KBr): 3407, 3300 (N-H), 2965, 2884 (C-H_._), 1601, 1497 (C=N, C=C_Ar_), 1290, 1135 (SO_2_) cm^−^^1^; ^1^H NMR (500 MHz, DMSO-*d*_6_) δ: 2.31 (s, 3H, CH_3_), 3.96 (s, 2H, S-CH_2_), 6.93–7.26 (m, 9H, H_Ar_ and 1H, H-3 and 2H, NH_2_), 8.02 (m, 1H, H-6), 9.53 (m, 1H, PhN**H**), 11.94 (m, 2H, NH, benzimidazolidine) ppm; Anal. calcd. for C_24_H_21_ClN_8_O_2_S_2_ (553.06); C, 52.12; H, 3.83; N, 20.26. Found: C, 51.78; H, 3.62; N, 19.75. HRMS (ESI-TOF) 552.0917 calcd for C_24_H_21_ClN_8_O_2_S_2_ [M + H]^+^ 553.0995 found 553.0976.

*2-[{4-Amino-6-[(4-fluorophenyl)amino]-1,3,5-triazin-2-yl}metylthio]-N-(1H-benzo[d]imidazol-2(3H)-ylidene)-4-chloro-5-methylbenzenesulfonamide* (**12**).

Starting from 1-(4-fluorophenyl)biguanide hydrochloride (0.371 g, 1.60 mmol). The title compound was obtained. Yield 0.091 g (20%); m.p. 278–290 °C with (dec.); IR (KBr): 3328, 3177 (N-H), 2967, 2886 (C-H), 1604, 1561, 1507, 1475 (C=N, C=C_Ar_), 1280, 1144 (SO_2_) cm^−^^1^; ^1^H NMR (500 MHz, DMSO-*d*_6_) δ: 2.31 (m, 3H, CH_3_), 3.96 (m, 2H, S-CH_2_), 7.05–7.73 (m, 8H, H_Ar_ and 1H, NH_2_ and 1H, H-3), 8.02 (m, 1H, H-6), 9.58 (m, 1H, NH, 4-F-C_6_H_5_-N**H**), 11.94 (m, 2H, NH, benzimidazolidine) ppm; Anal. calcd. for C_24_H_20_ClFN_8_O_2_S_2_ (571.05); C, 50.48; H, 3.53; N, 19.62. Found: C, 50.40; H, 3.50; N, 19,57. HRMS (ESI-TOF) (570.0823) calcd for C_24_H_20_ClFN_8_O_2_S_2_ [M + H]^+^ (571.0901) found 571.0902.

*2-[{4-Amino-6-[(4-chlorophenyl)amino]-1,3,5-triazin-2-yl}metylthio]-N-(1H-benzo[d]imidazol-2(3H)-ylidene)-4-chloro-5-methylbenzenesulfonamide* (**13**).

Starting from 1-(4-chlorophenyl)biguanide hydrochloride (0.397 g, 1.60 mmol). The resulting reaction mixture was treated with ethanol (2.5 mL) and precipitated solid was filtered off, then mixed with water (5 mL), filtered off, and dried. Yield 0.062 g (12%); m.p. 285–286 °C; IR (KBr): 3335, 3180 (N-H), 2956, 2888, 2787, 2681 (C-H), 1556, 1492 (C=N, C=C_Ar_), 1410, 1143 (SO_2_) cm^−^^1^; ^1^H NMR (500 MHz, DMSO-*d*_6_) δ: 2.31 (m, 3H, CH_3_), 3.98 (m, 2H, S-CH_2_), 7.09–8.02 (m, 8H, H_Ar_ and 2H, NH_2_ and 1H, H-3), 9.57 (m, 1H, H-6), 9.68 (m, 1H, NH 4-Cl-C_6_H_4_-N**H**), 11.93 (m, 2H, NH, benzimidazolidine) ppm; Anal. calcd. for C_24_H_20_Cl_2_N_8_O_2_S_2_ (587.50); C, 49.06; H, 3.43; N, 19.07. Found: C, 48.99; H, 3.38; N, 19.08. HRMS (ESI-TOF) (586.0528) calcd for C_24_H_20_Cl_2_N_8_O_2_S_2_ [M + H]^+^ (587.0606) found 587.0616.

*2-[{4-Amino-6-[(4-methoxyphenyl)amino]-1,3,5-triazin-2-yl}metylthio]-N-(1H-benzo[d]imidazol-2(3H)-ylidene)-4-chloro-5-methylbenzenesulfonamide* (**14**).

Starting from 1-(4-methoxyphenyl)biguanide hydrochloride (0.390 g, 1.60 mmol). The title compound was obtained after extraction of the impurities with boiling ethanol (1:215) and the precipitate was washed with acetonitrile (6 × 0.5 mL). Yield 0.067 g (14%); m.p. 278–290 °C (dec.); IR (KBr): 3350, 3319, 3176 (N-H), 2996, 2949, 2832 (C-H), 1558, 1473 (C=N, C=C_Ar_), 1282, 1141 (SO_2_) cm^−^^1^; ^1^H NMR (500 MHz, DMSO-*d*_6_) δ: 2.32 (s, 3H, CH_3_Ph), 3.70 (s, 3H, 4-MeO-C_6_H_4_-NH), 3.94 (s, 2H, S-CH_2_), 6.81–7.61 (m, 8H, H_Ar_ and 2H, NH_2_ and 1H, H-3), 8.02 (m, 1H, H-6), 9.38 (m, 1H, NH, 4-MeO-C_6_H_4_-N**H**), 11.95 (m, 2H, NH, benzimidazolidine) ppm; Anal. calcd. for C_25_H_23_ClN_8_O_3_S_2_ (583.08); C, 51.50; H, 3.98; N, 19.22. Found: C, 51,78; H, 4.05; N,19.56.

*2-{[4-Amino-6-(benzylamino)-1,3,5-triazin-2-yl]metylthio}-N-(1H-benzo[d]imidazol-2(3H)-ylidene)-4-chloro-5-methylbenzenesulfonamide* (**15**).

Starting from 1-benzylbiguanide hydrochloride (0.364 g, 1.60 mmol). The resulting reaction mixture was treated with ethanol (2 mL) and acetonitrile (5 mL). Precipitated solid was filtered off, then mixed with water (5 mL), filtered off, and dried. Yield 0.121 g (27%); m.p. 260.5–262.8 °C; IR (KBr): 3388, 3299, 3170 (N-H), 3030 (C-H_Ar_), 2965, 2941, 2859 (C-H), 1572, 1475 (C=N, C=C_Ar_), 1283, 1169 (SO_2_) cm^−^^1^; ^1^H NMR (500 MHz, DMSO-*d*_6_, T=100°C) δ: 2.33 (m, 3H, CH_3_), 3.86 (m, 2H, S-CH_2_), 4.48 (s, 2H, **CH_2_**-Ph), 6.44–7.29 (m, 9H, H_Ar_ and 2H, NH_2_), 7.73 (m, 1H, H-3), 7.98 (m, 1H, H-6), 11.64 (m, 2H, NH benzimidazole)ppm; Anal. calcd. for C_25_H_23_ClN_8_O_2_S_2_ (567.09); C, 52.95; H, 4.09; N, 19.76. Found: C, 52.89; H, 4.02; N, 19.57. HRMS (ESI-TOF) (566.1074) calcd for C_25_H_23_ClN_8_O_2_S_2_ [M+H]^+^ (567.1152) found 567.1146.

*2-[{4-Amino-6-[methyl(phenyl)amino]-1,3,5-triazin-2-yl}metylthio]-N-(1H-benzo[d]imidazol-2(3H)-ylidene)-4-chloro-5-methylbenzenesulfonamide* (**16**).

Starting from 1-phenyl-1-methylbiguanide hydrochloride (0.364 g, 1.60 mmol). The title compound was obtained after crystallization from a mixture of ethanol/acetonitrile (2:3). Yield 0.060g (13%); m.p. 231–232 °C; IR (KBr): 3352, 3324, 3224 (N-H), 2965, 2888 (C-H), 1604, 1562 (C=N, C=C_Ar_), 1291, 1140 (SO_2_) cm^−^^1^; ^1^H NMR (500 MHz, DMSO-*d*_6_) δ: 2.32 (s, 3H, CH_3_Ph), 3.88 (s, 2H, S-CH_2_), 6.92−7.35 (m, 10H, H_Ar_ and 2H, NH_2_), 7.80 (m, 1H, H-3), 8.00 (m, 1H, H-6), 11.94 (m, 2H, NH, benzimidazolidine) ppm; Anal. calcd. for C_25_H_23_ClN_8_O_2_S_2_ (567.09); C, 52.95; H, 4.09; N, 19.76. Found: C, 52.05; H, 4.02; N, 19.11. HRMS (ESI-TOF) 566.1074 calcd for C_25_H_23_ClN_8_O_2_S_2_ [M + H]^+^ 567.1152 found 567.1128.

*2-[{4-Amino-6-[(4-chlorophenyl)(methyl)amino]-1,3,5-triazin-2-yl}methylthio]-N-(1H-benzo[d]imidazol-2(3H)-ylidene)-4-chloro-5-methylbenzenesulfonamide* (**17**).

Starting from 1-(4-chlorophenyl)-1-methylbiguanide hydrochloride (0.418 g, 1.60 mmol)**.** The title compound was obtained after extraction of the impurities with hot ethanol (1:22) Yield 0.178 g (37%); m.p. 248–249 °C; IR (KBr): 3338, 3209 (N-H), 2962, 2920, 2882, 2851 (C-H), 1601, 1595, (C=N, C=C_Ar_), 1290, 1140 (SO_2_) cm^−1^; ^1^H NMR (500 MHz, DMSO-*d*_6_) δ: 2.32 (s, 3H, CH_3_), 3.31 (s, 3H, N-C**H**_3_), 3.88 (s, 2H, S-CH_2_), 6.99–7.77 (m, 8H, H_Ar_ and 2H, NH_2_), 7.77(m, 1H, H-3), 7.99 (m, 1H, H-6), 11.94 (m, 2H, NH, benzimidazolidine) ppm; Anal. calcd. for C_25_H_22_Cl_2_N_8_O_2_S_2_ (601.53); C, 49.92; H, 3.69; N, 18.63. Found: C, 49.51; H, 3.80; N, 18.32. HRMS (ESI-TOF) 504.0917 calcd for C_20_H_21_ClN_8_O_2_S_2_ [M + H]^+^ 505.0995 found 505.0983.

*2-{[4-Amino-6-(4-methylpiperazin-1-yl)-1,3,5-triazin-2-yl]metylthio}-N-(1H-benzo[d]imidazol-2(3H)-ylidene)-4-chloro-5-methylbenzenesulfonamide* (**18**).

Starting from *N*-carbamimidoyl-4-methylpiperazine-1-carboximidamide hydrochloride (0.352 g, 1.60 mmol). The resulting reaction mixture was treated with ethanol (2.5 mL) and precipitated solid was filtered off, then mixed with water (5 mL) filtered off, and dried. Yield 0.148 g (33%); m.p. 267–268 °C. IR (KBr): 3332, 3168 (NH), 3004 (C-H_Ar_), 2923, 2853 (C-H), 1593, 1475 (C=N, C=C_Ar_), 1276, 1141 (SO_2_) cm^−1^; ^1^H NMR (500 MHz, DMSO-*d*_6_) δ: 2.15 (s, 3H, C**H**_3_N), 2.21–2.24 (m, 4H, piperazine), 2.32 (s, 3H, CH_3_Ph), 3.62–3.64 (m, 4H, piperazine), 3.86 (s, 2H, S-CH_2_), 6.88–7.28 (m, 4H, H_Ar_ and 2H, NH_2_), 7.93 (m, 1H, H-3), 7.99 (m, 1H, H-6), 11.90 (m, 2H, NH, benzimidazolidine) ppm; Anal. calcd. for C_23_H_26_ClN_9_O_2_S_2_ (560.09); C, 49.32; H, 4.68; N, 22.51. Found: C, 49.37; H, 4.80; N, 21.49. HRMS (ESI-TOF) (559.1339) calcd. for C_23_H_26_ClN_9_O_2_S_2_ [M + H]^+^ (560.1417) found 560.1404.

*2-{[4-Amino-6-(4-phenylpiperazin-1-yl)-1,3,5-triazin-2-yl]methylthio}-N-(1H-benzo[d]imidazol-2(3H)-ylidene)-4-chloro-5-methylbenzenesulfonamide* (**19**).

Starting from *N*-carbamimidoyl-4-phenylpiperazine-1-carboximidamide hydrochloride (0.452 g, 1.60 mmol). The title compound was obtained after extraction of the impurities with boiling ethanol (1:11), a second fraction of the solid crystallized from filtrate. Yield 0.220 g (44%); m.p. 253–255 °C (dec.); IR (KBr): 3473, 3307 (N-H), 2955, 2923, 2860 (C-H), 1579, 1464 (C=N, C=C_Ar_), 1290, 1139 (SO_2_) cm^−^^1^; ^1^H NMR (500 MHz, DMSO-*d*_6_) δ: 2.30 (s, 3H, CH_3_), 3.06–3.11 (m, 4H, piperazine), 3.78–3.80 (m, 4H, piperazine), 3.88 (s, 2H, S-CH_2_), 6.80–7.22 (m, 9H, H_Ar_ and 2H, NH_2_), 7.89 (m, 1H, H-3), 7.98 (m, 1H, H-6), 11.83 (m, 1H, NH, benzimidazolidine) ppm; ^13^C NMR (DMSO-*d*_6_) δ: 19.40, 42.82, 48.72, 111.47, 116.38, 119.76, 122.20, 127.94, 129.43, 131.02, 131.35, 131.80, 136.61, 136.66, 139.84, 151.16, 151.39, 164.60, 167.36, 174.40 ppm; Anal. calcd. for C_28_H_28_ClN_9_O_2_S_2_ (622.16); C, 54.05; H, 4.54; N, 20.26. Found: C, 53.97; H, 4.73; N, 19.76. HRMS (ESI-TOF) 621.1496 calcd for C_28_H_28_ClN_9_O_2_S_2_ [M + H]^+^ 622.1574 found 622.1560.

*2-[{4-Amino-6-[4-(4-fluorophenyl)piperazin-1-yl]-1,3,5-triazin-2-yl}methylthio]-N-(1H-benzo[d]imidazol-2(3H)-ylidene)-4-chloro-5-methylbenzenesulfonamide* (**20**).

Starting from *N*-carbamimidoyl-4-(4-fluorophenyl)piperazine-1-carboximidamide hydrochloride (0.481 g, 1.60 mmol). The title compound was obtained after crystallization from ethanol (1:21). Yield 0.145 g (30%); m.p. 253–254 °C (dec.); IR (KBr): 3324, 3181 (NH), 2953, 2922 (C-H_._), 1582, 1448 (C = N, C = C_Ar_), 1289, 1139 (SO_2_) cm^−^^1^; ^1^H NMR (500 MHz, DMSO-*d*_6_) δ: 2.30 (s, 3H, CH_3_), 3.01 (m, 4H, piperazine), 3.77–3.79 (m, 4H, piperazine), 3.89 (s, 2H, S-CH_2_), 6.93–7.26 (m, 8H, H_Ar_ and 2H, NH_2_), 7.91 (m, 1H, H-3), 7.99 (m, 1H, H-6), 11.89 (m, 2H, NH, benzimidazolidine) ppm; Anal. calcd. for C_28_H_27_ClFN_9_O_2_S_2_ (640.15); C, 52.53; H, 4.25; N, 16.69. Found: C, 52.00; H, 4.11; N, 19.27. HRMS (ESI-TOF) 639.1402 calcd for C_28_H_27_ClFN_9_O_2_S_2_[M+H]^+^ 640.1480 found 640.1471.

*2-[{4-Amino-6-[4-(2-fluorophenyl)piperazin-1-yl]-1,3,5-triazin-2-yl}metylthio]-N-(1H-benzo[d]imidazol-2(3H)-ylidene)-4-chloro-5-methylbenzenesulfonamide* (**21**).

Starting from *N*-carbamimidoyl-4-(2-fluorophenyl)piperazine-1-carboximidamide hydrochloride (0.481 g, 1.60 mmol). The title compound was obtained after extraction of the impurities with boiling ethanol (1:20). Yield 0.178 g (34%); m.p. 280–282 °C with (dec.); IR (KBr): 3295, 3207, 3170 (N-H), 2944, 2888, 2853, 2822 (C-H), 1595, 1565, 1522, 1473 (C=N, C=C_Ar_), 1278, 1138 (SO_2_) cm^−^^1^; ^1^H NMR (500 MHz, DMSO-*d*_6_) δ: 2.31 (m, 3H, CH_3_), 2.92–2.96 (m, 4H, piperazine), 3.79–3.81 (m, 4H, piperazine), 3.89 (m, 2H, S-CH_2_), 6.96–7.28 (m, 8H, H_Ar_ and 2H, NH_2_), 7.92 (m, 1H, H-3), 8.00 (m, 1H, H-6), 11.95 (m, 2H, NH, benzimidazolidine) ppm; Anal. calcd. for C_28_H_27_ClFN_9_O_2_S_2_ (640.15); C, 52.53; H, 4.25; N, 19.69. Found: C, 52.01; H, 4.12; N, 19.36.

*2-{[4-Amino-6-{4-[4-(trifluoromethyl)phenyl]piperazin-1-yl}-1,3,5-triazin-2-yl]methylthio}-N-(1H-benzo[d]imidazol-2(3H)-ylidene)-4-chloro-5-methylbenzenesulfonamide* (**22**).

Starting from *N*-carbamimidoyl-4-[4-(trifluoromethyl)phenyl]piperazine-1-carboximidamide hydrochloride (0.561 g; 1.60 mmol). The title compound was obtained after extraction of the impurities with boiling ethanol (1:5). Yield 0.217 g (39%); m.p. 278.7–279.7 °C (dec.); IR (KBr): 3370, 3320, 3208 (N-H), 2924, 2895, 2854 (C-H), 1522, 1472 (C=N, C=C_Ar_), 1334, 1138 (SO_2_) cm^−^^1^; ^1^H NMR (500 MHz, DMSO-*d*_6_) δ: 2.29 (s, 3H, CH_3_), 3.28 (m, 4H, piperazine), 3.79–3.80 (m, 4H, piperazine), 3.87 (s, 2H, S-CH_2_), 6.96- 7.23 (m, 8H, H_Ar_ and 2H, NH_2_), 7.87 (m, 1H, H-3), 7.97 (m, 1H, H-6), 11.66 (m, 2H, NH, benzimidazolidine) ppm; Anal. calcd. for C_29_H_27_ClF_3_N_9_O_2_S_2_ (690.16); C, 50.47; H, 3.94; N, 18.27. Found: C, 50.41; H, 3.85; N, 18.22. HRMS (ESI-TOF) 689.1370 calcd for C_29_H_27_ClF_3_N_9_O_2_S_2_ [M+H]^+^ 690.1448 found 690.1452.

*2-[{4-Amino-6-[4-(4-chlorophenyl)piperazin-1-yl]-1,3,5-triazin-2-yl}methylthio]-N-(1H-benzo[d]imidazol-2(3H)-ylidene)-4-chloro-5-methylbenzenesulfonamide* (**23**).

Starting from *N*-carbamimidoyl-4-(4-chlorophenyl)piperazine-1-carboximidamide hydrochloride (0.508 g, 1.60 mmol). The title compound was obtained after crystallization from ethyl acetate (1:6), next to obtained oil Et_2_O (3 × 2 mL) was added, then obtained solid was purified by extraction of the impurities with boiling ethyl acetate (1:30). Yield 0.135 g (24%); m.p. 240–242 °C; IR (KBr): 3393, 3370, 3228 (N-H), 2980, 2916, 2857, 2833 (C-H), 1525 (C=N, C=C_Ar_), 1234, 1141 (SO_2_) cm^−^^1^; ^1^H NMR (500 MHz, DMSO-*d*_6_) δ: 2.25 (s, 3H, CH_3_), 3.13–3.14 (m, 4H, piperazine), 3.81–3.81 (m, 2H, S-CH_2_ and 4H, piperazine), 6.72–7.25 (m, 8H, H_Ar_ and 2H, NH_2_), 7.77 (m, 1H, H-3), 7.91 (m, 1H, H-6), 10.54 (m, 1H, NH, benzimidazolidine) ppm; Anal. calcd. for C_28_H_27_Cl_2_N_9_O_2_S_2_ (656.61); C, 51.22; H, 4.14; N, 19.20. Found: C, 51.20; H, 4.10; N, 19.17. HRMS (ESI-TOF) 655.1106 calcd for C_28_H_27_Cl_2_N_9_O_2_S_2_ [M+H]^+^ 656.1175 found 656.1175.

*2-[{4-Amino-6-[4-(3-chlorophenyl)piperazin-1-yl]-1,3,5-triazin-2-yl}methylthio]-N-(1H-benzo[d]imidazol-2(3H)-ylidene)-4-chloro-5-methylbenzenesulfonamide* (**24**).

Starting from *N*-carbamimidoyl-4-(3-chlorophenyl)piperazine-1-carboximidamide hydrochloride (0.508 g, 1.60 mmol). The title compound was obtained after extraction of the impurities with boiling ethanol (1:31) and the precipitate was washed with ethanol (2 × 2 mL). Yield 0.269g (51%); m.p. 239.5–240.4 °C. IR (KBr): 3374, 3289 (N-H), 2955, 2922, 2851, 2821 (C-H), 1594, 1567, 1523, 1488 (C=N, C=C_Ar_), 1278, 1136 (SO_2_) cm^−^^1^; ^1^H NMR (500 MHz, DMSO-*d*_6_) δ: 2.30 (s, 3H, CH_3_), 3.16 (m, 4H, piperazine), 3.79 (m, 4H, piperazine), 3.87 (s, 2H, S-CH_2_), 6.80–7.25 (m, 8H, H_Ar_ and 2H, NH_2_), 7.87 (m, 1H, H-3), 7.97 (m, 1H, H-6), 11.59 (m, 1H, NH, benzimidazolidine) ppm; Anal. calcd. for C_28_H_27_Cl_2_N_9_O_2_S_2_ (656.61) C, 51.22; H, 4.14; N, 19.20. Found: C, 51.47; H,4.22; N, 19.45.

*2-[{4-Amino-6-[4-(3,4-dichlorophenyl)piperazin-1-yl]-1,3,5-triazin-2-yl}metylthio]-N-(1H-benzo[d]imidazol-2(3H)-ylidene)-4-chloro-5-methylbenzenesulfonamide* (**25**).

Starting from *N*-carbamimidoyl-4-(3,4-dichlorophenyl)piperazine-1-carboximidamide hydrochloride (0.563 g, 1.60 mmol). The title compound was obtained after extraction of the impurities with boiling ethanol (1:200) a second fraction of the solid crystallized from filtrate. Yield 0.149 g (27%); m.p. 273–274 °C; IR (KBr): 3450, 3373, 3301 (N-H), 2927, 2834 (C-H), 1595, 1463 (C=N, C=C_Ar_), 1236, 1138 (SO_2_) cm^−^^1^; ^1^H NMR (500 MHz, DMSO-*d*_6_) δ: 2.31 (s, 3H, CH_3_), 3.14 (m, 4H, piperazine), 3.72–3.77 (m, 4H, piperazine), 3.89 (s, 2H, S-CH_2_), 6.93–7.42 (m, 7H, H_Ar_ and 2H, NH_2_), 7.92 (m, 1H, H-3), 8.00 (m, 1H, H-6), 11.93 (m, 2H, NH, benzimidazolidine) ppm; Anal. calcd. for C_28_H_26_Cl_3_N_9_O_2_S_2_ (691.05); C, 48.66; H, 3.79; N, 18.24. Found: C, 48.63; H, 3.67; N, 18.20. HRMS (ESI-TOF) (689.0716) calcd for C_28_H_26_Cl_3_N_9_O_2_S_2_ [M+H]^+^ (690.0794) found 690.0826.

*2-[{4-Amino-6-[4-(3-chloro-4-fluorophenyl)piperazin-1-yl]-1,3,5-triazin-2-yl}methylthio]-N-(1H-benzo[d]imidazol-2(3H)-ylidene)-4-chloro-5-methylbenzenesulfonamide* (**26**).

Starting from *N*-carbamimidoyl-4-(3-chloro-4-fluorophenyl)piperazine-1-carboximidamide hydrochloride (0.536 g, 1.60 mmol). The title compound was obtained after extraction of the impurities with boiling ethanol (1:38) and the precipitate was washed with 2 × 2 mL ethanol. Yield 0.220g (41%); m.p. 246.4–247.1 °C; IR (KBr): 3372, 3293, 3115 (N-H), 2955, 2924, 2859, 2835 (C-H), 1593, 1564, 1505, 1462 (C=N, C=C_Ar_), 1235, 1136 (SO_2_); ^1^H NMR (500 MHz, DMSO-*d*_6_) δ: 2.30 (s, 3H, CH_3_Ph), 3.09 (m, 4H, piperazine), 3.79 (m, 4H, piperazine), 3.87 (s, 2H, S-CH_2_), 6.94–7.28 (m, 7H, H_Ar_ and 2H, NH_2_), 7.88 (m, 1H, H-3), 7.98 (m, 1H, H-6), 11.00–12.0 (m, 2H, NH, benzimidazolidine) ppm Anal. calcd. for C_28_H_26_Cl_2_FN_9_O_2_S_2_ (674.60) C, 49.85; H, 3.88; N, 18.69. Found: C, 50.15; H, 4.08; N, 19.05.

*2-[{4-Amino-6-[4-(4-methoxyphenyl)piperazin-1-yl]-1,3,5-triazin-2-yl}metylthio]-N-(1H-benzo[d]imidazol-2(3H)-ylidene)-4-chloro-5-methylbenzenesulfonamide* (**27**).

Starting from *N*-carbamimidoyl-4-(2-methoxyphenyl)piperazine-1-carboximidamide hydrochloride (0.500 g, 1.60 mmol). The title compound was obtained after extraction of the impurities with boiling ethanol (1:50), a second fraction of the solid crystallized from filtrate. Then obtained crude solid was extracted again with boiling acetonitrile (1:50), a second fraction of the solid crystallized from filtrate. Yield 0.185 g (36%); m.p. 249–250 °C (dec.); IR (KBr): 3464, 3367, 3305 (N-H), 2954, 2856, 2831, 2813 (C-H), 1633 (NH_2def._), 1560, 1474 (C=N, C=C_Ar_), 1301, 1137 (SO_2_) cm^−^^1^; ^1^H NMR (500 MHz, DMSO-*d*_6_) δ: 2.29 (s, 3H, CH_3_Ph), 2.95 (m, 4H, piperazine), 3.68 (s, 3H, -OMe), 3.80 (m, 4H, piperazine), 3.86 (s, 2H, S-CH_2_), 6.82–7.18 (m, 11H, 8H, H_Ar_ and 2H, NH_2_ and 1H, NH), 7.86 (m, 1H, H-3), 7.97 (m, 1H, H-6) ppm; Anal. calcd. for C_29_H_30_ClN_9_O_3_S_2_ (652.19); C, 53.41; H, 4.64; N, 19.33. Found: C, 53.40; H, 4.60; N, 19.31. HRMS (ESI-TOF) (651.1602) calcd for C_29_H_30_ClN_9_O_3_S_2_ [M + H]^+^ (652.1680) found 652.1673.

*2-[{4-Amino-6-[4-(2-methoxyphenyl)piperazin-1-yl]-1,3,5-triazin-2-yl}metylthio]-N-(1H-benzo[d]imidazol-2(3H)-ylidene)-4-chloro-5-methylbenzenesulfonamide* (**28**).

Starting from *N*-carbamimidoyl-4-(2-methoxyphenyl)piperazine-1-carboximidamide hydrochloride (0.500 g, 1.60 mmol). The title compound was obtained after extraction of the impurities with boiling ethanol (1:10). Yield 0.207 g (40%); m.p. 263–264 °C (dec.); IR (KBr): 3373, 3287 (N-H), 2955, 2889, 2851 (C-H), 1564, 1465 (C=N, C=C_Ar_), 1343, 1137 (SO_2_), 1277 (Ar-O-C) cm^−^^1^; ^1^H NMR (500 MHz, DMSO-*d*_6_) δ: 2.31 (s, 3H, CH_3_), 2.88 (m, 4H, piperazine), 3.77- 3.79 (m, 4H, piperazine), 3.80 (s, 3H, O-C**H**_3_), 3.89 (s, 2H, S-CH_2_), 6.87–7.27 (m, 8H, H_Ar_ and 2H, NH_2_), 7.93 (m, 1H, H-3), 8.00 (m, 1H, H-6), 11.94 (m, 2H, NH, benzimidazolidine) ppm; Anal. calcd. for C_29_H_30_ClN_9_O_3_S_2_ (651.16); C, 53.41; H, 4.64; N, 19.33 Found: C, 53.38; H, 4.63; N, 19.30. HRMS (ESI-TOF) 651.1602 calcd for C_29_H_30_ClN_9_O_3_S_2_ [M+H]^+^ 652.1680 found 652.1671.

*2-[{4-Amino-6-[4-(4-nitrophenyl)piperazin-1-yl]-1,3,5-triazin-2-yl}metylthio]-N-(1H-benzo[d]imidazol-2(3H)-ylidene)-4-chloro-5-methylbenzenesulfonamide* (**29**).

Starting from *N*-carbamimidoyl-4-(4-nitrophenyl)piperazine-1-carboximidamide hydrochloride (0.524 g, 1.60mmol). The title compound was obtained after extraction of the impurities with boiling ethanol (1:80), remained part by crystallization from filtrate. Yield 0.185 g (35%); m.p. 213–214 °C (dec.); IR (KBr): 3373, 3329 (N-H), 2921, 2896, 2860 (C-H), 1599, 1474 (C=N, C=C_Ar_), 1316, 1132 (SO_2_) cm^-1^; ^1^H NMR (500 MHz, DMSO-*d*_6_) δ: 2.27 (s, 3H, CH_3_), 3.51 (m, 4H, piperazine), 3.84 (m, 4H, piperazine and 2H, S-CH_2_), 6.83–7.94 (m, 8H, H_Ar_), 8.06 (m, 1H, H-6), 8.08 (m, 1H, H-3), 11.00–12.00 (m, 2H, NH, benzimidazolidine) ppm; Anal. calcd. for C_28_H_27_ClN_10_O_4_S_2_ (667.16); C, 50.41; H, 4.08; N, 20.99. Found: C, 50.38; H, 4.04; N, 20.18. HRMS (ESI-TOF) 666.1347 calcd for C_28_H_27_ClN_10_O_4_S_2_ [M+H]^+^ 667.1425 found 667.1429.

*2-{[4-Amino-6-(4-benzylpiperazin-1-yl)-1,3,5-triazin-2-yl]methylthio}-N-(1H-benzo[d]imidazol-2(3H)-ylidene)-4-chloro-5-methylbenzenesulfonamide* (**30**).

Starting from 4-benzyl-*N*-carbamimidoylpiperazine-1-carboximidamide hydrochloride (0.475 g, 1.60 mmol). The title compound was obtained after extraction of the impurities with boiling ethanol (1:21) and the precipitate was washed with ethanol (2 × 2 mL). Yield 0.183g (36%); m.p. 244–245 °C; IR (KBr): 3367, 3316 (N-H), 2934, 2919, 2890, 2856 (C-H), 1562 (C=N, C=C_Ar_), 1279, 1137 (SO_2_) cm^−^^1^; ^1^H NMR (500 MHz, DMSO-*d*_6_) δ: 2.28–2.31 (m, 3H, CH_3_ and 4H, piperazine), 3.45 (s, 2H, C**H**_2_Ph), 3.62–3.64 (m, 4H, piperazine), 3.86 (m, 2H, S-CH_2_), 6.90–7.35 (m, 9H, H_Ar_ and 2H, NH_2_), 7.91 (m, 1H, H-3), 7.99 (m, 1H, H-6), 11.92 (m, 2H, NH, benzimidazolidine) ppm; ^13^C NMR (DMSO-*d*_6_) δ: 19.39, 40.55, 42.84, 52.60, 62.42, 111.38, 122.87, 127.47, 128.06, 128.67, 129.32, 129.84, 130.85, 131.94, 136.71, 136.92, 138.41, 139.25, 150.19, 164.53, 167.30, 174.24 ppm; Anal. calcd. for C_29_H_30_ClN_9_O_2_S_2_ (636.19); C, 54.75; H, 4.75; N, 19.81. Found: C, 54.69; H, 4.62; N, 19.76.

*2-[{4-Amino-6-[4-(4-benzhydrylphenyl)piperazin-1-yl]-1,3,5-triazin-2-yl}metylthio]-N-(1H-benzo[d]imidazol-2(3H)-ylidene)-4-chloro-5-methylbenzenesulfonamide* (**31**).

Starting from (0.597 g, 1.60 mmol) 4-benzhydryl-*N*-carbamimidoylpiperazine-1-carboximidamide hydrochloride. The title compound was obtained after extraction of the impurities with boiling ethanol (1:22). Yield 0.280 g (44%); m.p. 246–248 °C (dec.); IR (KBr): 3397, 3232 (N-H), 2966, 2919, 2860 (C-H), 1571, 1542, 1519, 1476 (C=N, C=C_Ar_), 1247, 1139 (SO_2_) cm^−^^1^; ^1^H NMR (500 MHz, DMSO-*d*_6_) δ: 2.20–2.32 (m, 3H, CH_3_ and 4H, piperazine), 3.66 (m, 4H, piperazine), 3.79 (m, 2H, S-CH_2_), 4.29 (s, 1H, benzhydryl), 6.87–7.43 (m, 14H, H_Ar_ and 2H, NH_2_), 7.76 (m, 1H, H-3), 7.91 (m, 1H, H-6), 11.29 (m, 1H, NH, benzimidazolidine) ppm; ^13^C NMR (DMSO-d_6_) δ: 19.39, 40.55, 42.84, 52,60 62.42, 111.38, 122.87, 127.47, 128.06, 128.67, 129.32, 129.84, 130.85, 131.94, 136.71, 136.92, 138.41, 139.25, 150.19, 164.53, 167.30, 174.24 ppm; Anal. calcd. for C_35_H_34_ClN_9_O_2_S_2_ (712.29); C, 59.02; H, 4.81; N, 17.70. Found: C, 58.97; H, 4.71; N, 17.65.

*2-[{4-Amino-6-[4-(phenylsulfonyl)piperazin-1-yl]-1,3,5-triazin-2-yl}methylthio]-N-(1H-benzo[d]imidazol-2(3H)-ylidene)-4-chloro-5-methylbenzenesulfonamide* (**32**).

Starting from *N*-carbamimidoyl-4-(phenylsulfonyl)piperazine-1-carboximidamide hydrochloride (0.555 g, 1.60 mmol). The title compound was obtained after extraction of the impurities with boiling ethanol (1:38) and next with boiling acetonitrile (1:21), the precipitate was washed with ethanol (2 × 2 mL) and acetonitrile (2 × 2 mL). Yield 0.246g (45%); m.p. 235.7–236.4 °C; IR (KBr): 3386, 3314, 3211 (N-H), 2979, 2928, 2853 (C-H), 1521, 1474 (C=N, C=C_Ar_), 1311, 1286, 1167, 1142 (SO_2_); ^1^H NMR (500 MHz, DMSO-*d*_6_) δ: 2.31 (s, 3H, CH_3_Ph), 2.88–2.93 (br.s., 4H, piperazine), 3.73 (m, 4H, piperazine), 3.83 (s, 2H, S-CH_2_), 6.81–7.77 (m, 9 H_Ar_ and 2H, NH_2_, 1H, H-3, 1H NH), 7.98 (m, 1H, H-6), 11.82 (m, 1H, NH, benzimidazolidine) ppm; Anal. calcd. for C_28_H_28_ClN_9_O_4_S_3_ (686.23); C, 49.01; H, 4.11; N, 18.37. Found: C, 48.55; H, 4.20; N, 18.65. HRMS (ESI-TOF) (685.1115) calcd for C_28_H_28_ClN_9_O_4_S_3_ [M + H]^+^ (686.1193) found 686.1232.

#### 3.2.3. General Procedure for the Preparation of 6-Substituted (*E*)-2-{[4-amino-1,3,5-triazin-2-yl]metylthio}-4-chloro-*N*-(5-fluoro-1*H*-benzo[d]imidazol-2(3*H*)-ylidene)-5-methylbenzenesulfonamide **33**–**40**


To the solution of sodium methoxide prepared from sodium (0.0368 g, 1.60 mmol) and anhydrous methanol (7.5 mL), ethyl 2-[{5-chloro-2-[*N*-(5-fluoro-1*H*-benzo[*d*]imidazol-2(3*H*)-ylidene)sulfamoyl]-4-methylphenyl}thio]acetate (**4**) (0.366 g, 0.80 mmol) and appropriate biguanide hydrochloride (1.60 mmol) were added. The reaction mixture was stirred under reflux for 45 h. After cooling the precipitate was filtered off and dried, then stirred vigorously with water (25 mL) for 25 min. The crude product was purified by crystallization from an appropriate solvent or by extraction of the impurities with boiling ethanol.

*2-{[4-Amino-6-(3,5,5-trimethyl-4,5-dihydro-1H-pyrazol-1-yl)-1,3,5-triazin-2-yl]metylthio}-4-chloro-N-(5-fluoro-1H-benzo[d]imidazol-2(3H)-ylidene)-5-methylbenzenesulfonamide* (**33**).

Starting from *N*-carbamimidoyl-3,5,5-trimethyl-4,5-dihydro-1*H*-pyrazole-1-carboximidamide hydrochloride (0.372 g, 1.60 mmol). The title compound was obtained after crystallization from ethanol (1:14). Yield 0.169 g (35%); m.p. 239–241 °C; IR (KBr): 3346, 3385, 3223 (N-H), 2965, 2921, 2859 (C-H), 1530, 1474 (C=N, C=C_Ar_), 1336, 1132 (SO_2_) cm^−^^1^; ^1^H NMR (500 MHz, DMSO-*d*_6_) δ: 1.49–1.55 (m, 6H, C**H**_3_ pyrazole), 1.95–2.00 (m, 3H, C**H**_3_ pyrazole), 2.25–2.27 (s, 3H, CH_3_Ph), 2.78 (m, 2H, CH_2_, pyrazole), 3.86 (m, 2H, S-CH_2_), 6.56–7.57 (m, 3H, H_Ar_ and 2H, NH_2_, 1H, H-3), 7.91 (m, 1H, H-6), 10.81 (m, 1H, NH, 5-fluorobenzimidazolidine) ppm; Anal. calcd. For C_24_H_25_ClFN_9_O_2_S_2_ (590.10); C, 48.85; H, 4.27; N, 21.36. Found: C, 48.99; H, 4.19; N, 21.56.

*2-{[4-Amino-6-(indolin-1-yl)-1,3,5-triazin-2-yl]metylthio}-4-chloro-N-(5-fluoro-1H-benzo[d]imidazol-2(3H)-ylidene)-5-methylbenzenesulfonamide* (**34**).

Starting from *N*-carbamimidoylindoline-1-carboximidamide hydrochloride (0.384 g, 1.60 mmol). The title compound was obtained after extraction of the impurities with boiling ethanol (1:45), a second fraction of the solid crystallized from filtrate. Yield 0.120 g (25%); m.p. 267–269 °C; IR (KBr): 3420, 3324, 3186 (N-H), 2960, 2922, 2857 (C-H), 1515, 1481 (C=N, C=C_Ar_), 1385, 1133 (SO_2_) cm^−^^1^; ^1^H NMR (500 MHz, DMSO-*d*_6_) δ: 2.30 (s, 3H, CH_3_), 3.06–3.09 (t, *J=8.6 Hz*, 2H, 3*H*-indolinyl), 3.95–3.99 (m, 2H, S-CH_2_), 4.03–4.07 (t, 2H, *J=8.6 Hz*, 2H, 2*H*-indolinyl), 6.79–7.79 (m, 7H, 7 H_Ar_ and 2H, NH_2_, 1H, H-3), 7.97 (m, 1H, H-6), 11.80 (m, 2H, NH, 5-fluorobenzimidazolidine) ppm; Anal. calcd. for C_26_H_22_ClFN_8_O_2_S_2_ (597.09); C, 52.30; H, 3.71; N, 18.77. Found: C, 52.26; H, 3.70; N, 18.53. HRMS (ESI-TOF) (596.0980) calcd for C_26_H_22_ClFN_8_O_2_S_2_ [M + H]^+^ (597.1058) found 597.1050.

*2-[{4-Amino-6-[(3-chlorophenyl)amino]-1,3,5-triazin-2-yl}methylthio]-4-chloro-N-(5-fluoro-1H-benzo[d]imidazol-2(3H)-ylidene)-5-methylbenzenesulfonamide* (**35**).

Starting from 1-(3-chlorophenyl)biguanide hydrochloride (0.396 g, 1.60 mmol). The resulting reaction mixture was treated with ethanol (3 mL) and precipitated solid was filtered off, then mixed with water (5 mL), filtered off, and dried. Yield 0.053 g (11%); m.p. 288.4–289.5 °C; IR (KBr): 3395, 3191 (N-H), 2957, 2924, 2852 (C-H), 1574, 1532, 1501, 1450 (C=N, C=C_Ar_), 1296, 1140 (SO_2_) cm^−^^1^; ^1^H NMR (500 MHz, DMSO-*d*_6_) δ: 2.32 (s, 3H, CH_3_), 3.98 (s, 2H, S-CH_2_), 6.92–7.74 (m, 7H, H_Ar_ and 2H, NH_2_), 7.85 (m, 1H, H-3), 8.01 (m, 1H, H-6), 9.72 (m, 1H, NH), 12.00–12.05 (m, 2H, NH, 5-fluorobenzimidazolidine) ppm; Anal. calcd. for C_24_H_19_Cl_2_FN_8_O_2_S_22_ (605.49); C, 47.61; H, 3.16; N, 18.51. Found: C, 47.57; H, 3.44; N, 17.03. HRMS (ESI-TOF) (604.0433) calcd for C_24_H_19_Cl_2_FN_8_O_2_S_2_ [M + H]^+^ (605.0511) found 605.0551.

*2-[{4-Amino-6-[(4-chlorophenyl)(methyl)amino]-1,3,5-triazin-2-yl}metylthio]-4-chloro-N-(5-fluoro-1H-benzo[d]imidazol-2(3H)-ylidene)-5-methylbenzenesulfonamide* (**36**).

Starting from 1-(4-chlorophenyl)-1-methylbiguanide hydrochloride (0.418 g, 1.60 mmol). The title compound was obtained after crystallization from ethanol (1:20). Yield 0.074 g (15%); m.p. 271–272 °C; IR (KBr): 3300, 3180 (N-H), 2951, 2925, 2859 (C-H), 1581, 1571, 1528, 1492 (C=N, C=C_Ar_), 1294, 1139 (SO_2_) cm^−^^1^; ^1^H NMR (500 MHz, DMSO-*d*_6_) δ: 2.32 (m, 3H, CH_3_Ph), 3.32 (m, 3H, N-C**H**_3_), 3.88 (m, 2H, S-CH_2_), 6.93–7.36 (m, 7H, H_Ar_ and 2H, NH_2_), 7.77 (m, 1H, H-3), 7.97 (m, 1H, H-6), 12.01 (m, 2H, NH, 5-fluorobenzimidazolidine) ppm; Anal. calcd. for C_25_H_21_Cl_2_FN_8_O_2_S_2_ (619.52); C, 48.47; H, 3.42; N, 18.09. Found: C, 48.42; H, 3.16; N, 16.49. HRMS (ESI-TOF) 618.0590 calcd for C_25_H_21_Cl_2_FN_8_O_2_S_2_ [M + H]^+^ 619.0668 found 619.0662.

*2-[{[4-Amino-6-(4-phenylpiperazin-1-yl)-1,3,5-triazin-2-yl]methyl}thio]-4-chloro-N-(5-fluoro-1H-benzo[d]imidazol-2(3H)-ylidene)-5-methylbenzenesulfonamide* (**37**).

Starting from *N*-carbamimidoyl-4-phenylpiperazine-1-carboximidamide hydrochloride (0.452 g, 1.60 mmol). The title compound was obtained after extraction of the impurities with boiling ethanol (1: 2.5), a second fraction of the solid crystallized from filtrate. Yield 0.225 g (44%); m.p. 262–263 °C; IR (KBr): 3309, 3188, 3146 (N-H), 2957, 2921, 2895, 2860 (C-H), 1587, 1579, 1519, 1503 (C=N, C=C_Ar_), 1287, 1137 (SO_2_) cm^−^^1^; ^1^H NMR (500 MHz, DMSO-*d*_6_) δ: 2.28 (s, 3H, CH_3_), 3.10 (m, 4H, piperazine), 3.81 (m, 4H, piperazine), 3.84 (m, 2H, S-CH_2_), 6.78–7.24 (m, 8H, H_Ar_ and 2H, NH_2_), 7.85 (m, 1H, H-3), 7.93 (m, 1H, H-6), 11.49 (m, 1H, NH, 5-fluorobenzimidazolidine) ppm; Anal. calcd. for C_28_H_27_ClFN_9_O_2_S_2_ (640.15); C, 52.53; H, 4.25; N, 19.69. Found: C, 52.28; H, 4.08; N, 19.31. HRMS (ESI-TOF) 639.1402 calcd for C_28_H_27_ClFN_9_O_2_S_2_ [M + H]^+^ 640.1480 found 640.1471.

*2-[{4-Amino-6-[4-(4-fluorophenyl)piperazin-1-yl]-1,3,5-triazin-2-yl}metylthio]-4-chloro-N-(5-fluoro-1H-benzo[d]imidazol-2(3H)-ylidene)-5-methylbenzenesulfonamide* (**38**).

Starting from *N*-carbamimidoyl-4-(4-fluorophenyl)piperazine-1-carboximidamide hydrochloride (0.481 g, 1.60 mmol). The title compound was obtained after extraction of the impurities with boiling ethanol (1:5). Yield 0.104 g (20%); m.p. 261–262 °C (dec.); IR (KBr): 3477, 3337, 3302 (N-H), 2956, 2921, 2868 (C-H), 1589, 1567, 1511 (C=N, C=C_Ar_), 1292, 1137 (SO_2_) cm^−^^1^; ^1^H NMR (500 MHz, DMSO-*d*_6_) δ: 2.31 (s, 3H, CH_3_), 3.01 (m, 4H, piperazine), 3.78 (t, *J*=5.2 *Hz*, 4H, piperazine), 3.89 (s, 2H, S-CH_2_), 6.93–7.26 (m, 7H, H_Ar_ and 2H, NH_2_), 7.92 (m, 1H, H-3), 7.97 (m, 1H, H-6), 12.00 (m, 2H, NH, 5-fluorobenzimidazolidine) ppm; Anal. calcd. for C_28_H_26_ClF_2_N_9_O_2_S_2_ (658.14); C, 51.10; H, 3.98; N, 19.15. Found: C, 50.59; H, 3.92; N, 19.11. HRMS (ESI-TOF) 657.1307calcd for C_28_H_26_ClF_2_N_9_O_2_S_2_ [M + H]^+^ 658.1385 found 658.1385.

*2-[{4-Amino-6-[4-(4-chlorophenyl)piperazin-1-yl]-1,3,5-triazin-2-yl}metylthio]-4-chloro-N-(5-fluoro-1H-benzo[d]imidazol-2(3H)-ylidene)-5-methylbenzenesulfonamide* (**39**).

Starting from *N*-carbamimidoyl-4-(4-chlorophenyl)piperazine-1-carboximidamide hydrochloride (0.508 g, 1.60mmol). The title compound was obtained after extraction of the impurities with boiling ethanol (1:4.5), and a second fraction of the solid was crystallized from the filtrate. Yield 0.164 g (30%); m.p. 198–199 °C; IR (KBr): 3397, 3335 (N-H), 2922, 2894, 2856 (C-H), 1554, 1475 (C=N, C=C_Ar_), 1343, 1131 (SO_2_) cm^−^^1^; ^1^H NMR (500 MHz, DMSO-*d*_6_) δ: 2.25 (s, 3H, CH_3_), 3.12 (m, 4H, piperazine), 3.81–3.86 (m, 4H piperazine and 2H, S-CH_2_), 6.52–7.25 (m, 7H, H_Ar_ and 2H, NH_2_), 7.79 (m, 1H, H-3), 7.90 (m, 1H, H-6), 10.70 (m, 1H, NH, 5-fluorobenzimidazolidine) ppm; Anal. calcd. for C_28_H_26_Cl_2_FN_9_O_2_S_2_ (674.60); C, 49.85; H, 3.88; N, 18.69. Found: C, 49.45; H, 3.75; N, 18.39.

*2-[{4-Amino-6-[4-(3,4-dichlorophenyl)piperazin-1-yl]-1,3,5-triazin-2-yl}metylthio]-4-chloro-N-(5-fluoro-1H-benzo[d]imidazol-2(3H)-ylidene)-5-methylbenzenesulfonamide* (**40**).

Starting from *N*-carbamimidoyl-4-(3,4-dichlorophenyl)piperazine-1-carboximidamide hydrochloride (0.563 g, 1.60 mmol). To the resulting reaction mixture charcoal was added, the filtrate was evaporated to dryness, then mixed with water (5 mL), filtered off, and dried and then crystallized from acetonitrile. Yield 0.071 g (13%); m.p. 262–263 °C; IR (KBr): 3376, 3292, 3116 (N-H), 2920, 2858, 2833 (C-H), 1603, 1562, 1523, 1501 (C=N, C=C_Ar_), 1237, 1137 (SO_2_) cm^−^^1^; ^1^H NMR (500 MHz, DMSO-*d*_6_) δ: 2.31 (s, 3H, CH_3_), 3.16 (m, 4H, piperazine), 3.75–3.77 (m, 4H, piperazine), 3.89 (s, 2H, S-CH_2_), 6.93–7.42 (m, 6H, H_Ar_ and 2H, NH_2_), 7.93 (m, 1H, H-3), 7.97 (m, 1H, H-6), 12.00 (m, 2H, NH, 5-fluorobenzimidazolidine) ppm; Anal. calcd. for C_28_H_25_Cl_3_FN_9_O_2_S_2_ (709.05); C, 47.43; H, 3.55; N, 17.78. Found: C, 47.35; H, 3.50; N, 17.42. HRMS (ESI-TOF) (707.0622) calcd for C_28_H_25_Cl_3_FN_9_O_2_S_2_ [M+H]^+^ (708.0700) found. 708.0721.General Procedure for the Preparation of 6-Substituted (E)-2-{[4-Amino-1,3,5-triazin-2-yl]metylthio}-4-chloro-N-(5-chloro-1H-benzo[d]imidazol-2(3H)-ylidene)-5-methylbenzenesulfon-amide **41**–**49**.

To the solution of sodium methoxide prepared from sodium (0.0368 g, 1.60 mmol) and anhydrous methanol (7.5 mL) ethyl 2-[{5-chloro-2-[*N*-(5-chloro-1*H*-benzo[*d*]imidazol-2(3*H*)-ylidene)sulfamoyl]-4-methylphenyl}thio]acetate (**5**) (0.308 g, 0.80 mmol) and appropriate biguanide hydrochloride (1.60 mmol) were added. The reaction mixture was stirred under reflux for 45 h. After cooling, the precipitate was filtered off and dried, then stirred vigorously with water (25 mL) for 25 min. The crude product was purified by crystallization from the appropriate solvent or by extraction of the impurities with boiling ethanol.

*2-{[4-Amino-6-(3,5,5-trimethyl-4,5-dihydro-1H-pyrazol-1-yl)-1,3,5-triazin-2-yl]metylthio}-4-chloro-N-(5-chloro-1H-benzo[d]imidazol-2(3H)-ylidene)-5-methylbenzenesulfonamide* (**41**).

Starting from *N*-carbamimidoyl-3,5,5-trimethyl-4,5-dihydro-1*H*-pyrazole-1-carboximidamide hydrochloride (0.372 g, 1.60 mmol). The title compound was obtained after crystallization from a mixture of ethanol/acetonitrile (5:3). Yield 0.124 g (32%); m.p. 242–243 °C; IR (KBr): 3389, 3227 (N-H), 2965, 2921, 2860 (C-H), 1595, 1528, 1461 (C=N, C=C_Ar_), 1382, 1137 (SO_2_) cm^−^^1^; ^1^H NMR (500 MHz, DMSO-*d*_6_) δ: 1.50–1.54 (m, 6H, C**H**_3_ pyrazole), 1.96 -2.00 (m, 3H, C**H**_3_, pyrazole), 2.26 (m, 3H, CH_3_Ph), 2.78–2.80 (m, 2H, CH_2_, pyrazole), 3.86 (m, 2H, S-CH_2_), 6.74–7.57 (m, 3H, H_Ar_ and 2H, NH_2_ and 1H, H-3), 7.91 (m, 1H, H-6), 10.81 (m, 1H, NH, 5-chlorobenzimidazolidine) ppm; Anal. calcd. for C_24_H_25_Cl_2_N_9_O_2_S_2_ (605.10); C, 47.52; H, 4.15; N, 20.78. Found: C, 47.50; H, 4.11; N, 20.76. HRMS (ESI-TOF) 605.0950 calcd for C_24_H_25_Cl_2_N_9_O_2_S_2_ [M + H]^+^ 606.1028 found 606.1024.

*2-{[4-Amino-6-(phenylamino)-1,3,5-triazin-2-yl]methylthio}-4-chloro-N-(5-chloro-1H-benzo[d]imidazol-2(3H)-ylidene)-5-methylbenzenesulfonamide* (**42**).

Starting from 1-phenylbiguanide hydrochloride (0.342 g, 1.60 mmol). The title compound was obtained after extraction of the impurities with boiling ethanol (1:4.3), a second fraction of the solid crystallized from filtrate. Yield 0.139 g (37%); m.p. 160–162 °C; IR (KBr): 3388, 3334 (N-H), 2925, 2856 (C-H), 1597, 1575, 1531 (C=N, C=C_Ar_), 1272, 1133 (SO_2_) cm^−^^1^: ^1^H NMR (500 MHz, DMSO-*d*_6_) δ: 2.27 (s, 3H, CH_3_), 3.91 (m, 2H, S-CH_2_), 6.83–7.77 (m, 8H, H_Ar_ and 2H, NH_2_ and 1H, H-3), 7.94 (m, 1H, H-6), 10.89 (m, 1H, NH, 5-chlorobenzimidazolidine) ppm; Anal. calcd. for C_24_H_20_Cl_2_N_8_O_2_S_2_ (587.50); C, 49.06; H, 3.43; N, 19.07. Found: C, 49.15; H, 3.79; N, 19.05. HRMS (ESI-TOF) 586.0528 calcd for C_24_H_20_Cl_2_N_8_O_2_S_2_ [M + H]^+^ 587.0606 found 587.0606.

*2-[{4-Amino-6-[(4-fluorophenyl)amino]-1,3,5-triazin-2-yl}methylthio]-4-chloro-N-(5-chloro-1H-benzo[d]imidazol-2(3H)-ylidene)-5-methylbenzenesulfonamide* (**43**).

Starting from 1-(4-fluorophenyl)biguanide hydrochloride (0.371 g, 1.60 mmol). The resulting reaction mixture was treated with Et_2_O (30 mL) and the precipitated solid was filtered off, then mixed with water (15 mL), filtered off, and dried, and then crystallized from ethanol. Yield 0.073g (15%); m.p. 270.1–271.4 °C; IR (KBr): 3374, 3217 (N-H), 2961, 2922, 2856, 2832 (C-H), 1593, 1506 (C=N, C=C_Ar_), 1282, 1140 (SO_2_) cm^−^^1^; ^1^H NMR (500 MHz, DMSO-*d*_6_) δ: 2.32 (m, 3H, CH_3_), 3.97 (m, 2H, S-CH_2_), 7.05–8.00 (m, 7H, H_Ar_ and 2H, NH_2_), 7.61 (m, 1H, H-3), 8.00 (m, 1H, H-6), 9.56 (m, 1H, NH, 4-F-C_6_H_4_-N**H**), 12.06–12.10 (m, 2H, NH, 5-chlorobenzimidazolidine) ppm; Anal. calcd. for C_24_H_19_Cl_2_FN_8_O_2_S_2_ (605.49); C, 47.61; H, 3.16; N, 18.51. Found: C, 47.85; H, 3.23; N, 18.49. HRMS (ESI-TOF) (604.0433) calcd for C_24_H_19_Cl_2_FN_8_O_2_S_2_ [M + H]^+^ (605.0511) found 605.0505.

*2-[{4-Amino-6-[(4-methoxyphenyl)amino]-1,3,5-triazin-2-yl}methylthio]-4-chloro-N-(5-chloro-1H-benzo[d]imidazol-2(3H)-ylidene)-5-methylbenzenesulfonamide* (**44**).

Starting from 1-(4-methoxyphenyl)biguanide hydrochloride (0.390 g, 1.60 mmol). The title compound was obtained after extraction of the impurities with boiling acetonitrile (1:19), and a second fraction of the solid was crystallized from the filtrate. 0.051g (10%); m.p. 261–262 °C; IR (KBr): 3350, 3178 (N-H), 2947, 2929, 2831 (C-H), 1598, 1510 (C=N, C=C_Ar_), 1279, 1174 (SO_2_) cm^−^^1^; ^1^H NMR (500 MHz, DMSO-*d*_6_) δ: 2.29 (s, 3H, CH_3_Ph), 3.71 (s, 3H, OCH_3_), 3.94 (s, 2H, S-CH_2_), 6.81–7.61 (m, 7H, H_Ar_ and 2H, NH_2_ and 1H, H-3), 7.99 (m, 1H, H-6), 9.37 (m, 1H, NH, 4-MeO-C_6_H_4_-N**H**), 12.03 (m, 2H, NH, 5-chlorobenzimidazolidine) ppm; Anal. calcd. for C_25_H_22_Cl_2_N_8_O_3_S_2_ (617.53); C, 48.62; H, 3.59; N, 18.15. Found: C, 48.60; H, 3.55; N, 18.12.

*2-[{4-Amino-6-[4-(4-phenyl)piperazin-1-yl]-1,3,5-triazin-2-yl}metylthio]-4-chloro-N-(5-chloro-1H-benzo[d]imidazol-2(3H)-ylidene)-5-methylbenzenesulfonamide* (**45**).

Starting from *N*-carbamimidoyl-4-phenylpiperazine-1-carboximidamide hydrochloride (0.452 g, 1.60 mmol). The title compound was obtained after extraction of the impurities with boiling ethanol (1:4.5), a second fraction of the solid was crystallized from the filtrate. Yield 0.121g (30%); m.p. 217–219 °C; IR (KBr): 3357, 3313 (N-H), 2988, 2917, 2858, 2822 (C-H), 1586, 1485 (C=N, C=C_Ar_), 1290, 1139 (SO_2_) cm^−^^1^; ^1^H NMR (500 MHz, DMSO-*d*_6_) δ: 2.31 (s, 3H, -CH_3_), 3.05–3.10 (m, 4H, piperazine), 3.77–3.79 (m, 4H, piperazine), 3.89 (s, 2H, S-CH_2_), 6.79–7.24 (m, 8H, H_Ar_ and 2H, NH_2_), 7.91 (m, 1H, H-3), 7.97 (m, 1H, H-6), 11.99 (m, 2H, NH, 5-chlorobenzimidazolidine) ppm; Anal. calcd. for C_28_H_27_Cl_2_N_9_O_2_S_2_ (656.51); C, 51.22; H, 4.14; N, 19.20. Found: C, 51.18; H, 4.04; N, 18.44. HRMS (ESI-TOF) 655.1106 calcd for C_28_H_27_Cl_2_N_9_O_2_S_2_ [M+H]^+^ 656.1184 found 656.1182.

*2-[{4-Amino-6-[4-(4-fluorophenyl)piperazin-1-yl]-1,3,5-triazin-2-yl}metylthio]-4-chloro-N-(5-chloro-1H-benzo[d]imidazol-2(3H)-ylidene)-5-methylbenzenesulfonamide* (**46**).

Starting from *N*-carbamimidoyl-4-(4-fluorophenyl)piperazine-1-carboximidamide hydrochloride (0.481 g, 1.60 mmol). The title compound was obtained after extraction of the impurities with boiling ethanol (1:8). Yield 0.086 g (16%); m.p. 272–273 °C; IR (KBr): 3370, 3304, 3142 (N-H), 2955, 2900, 2867 (C-H), 1585, 1568, 1511, 1485 (C=N, C=C_Ar_), 1290, 1139 (SO_2_) cm^−^^1^; ^1^H NMR (500 MHz, DMSO-*d*_6_) δ: 2.31 (s, 3H, CH_3_), 3.01 (m, 4H, piperazine), 3.76–3.78 (m, 4H, piperazine), 3.89 (s, 2H, S-CH_2_), 6.94–7.29 (m, 7H, H_Ar_ and 2H, NH_2_), 7.93 (m, 1H, H-3), 7.97 (m, 1H, H-6), 12.06 (m, 2H, NH, 5-chlorobenzimidazolidine) ppm; Anal. calcd. for C_28_H_26_Cl_2_N_9_O_2_S_2_ (674.60); C, 49.85; H, 3.88; N, 18.69. Found: C, 49.56; H, 3.62; N, 18.19.

*2-{[4-Amino-6-{4-[4-(trifluoromethyl)phenyl]piperazin-1-yl}-1,3,5-triazin-2-yl]methylthio}-4-chloro-N-(5-chloro-1H-benzo[d]imidazol-2(3H)-ylidene)-5-methylbenzenesulfonamide* (**47**).

Starting from *N*-carbamimidoyl-4-[4-(trifluoromethyl)phenyl]piperazine-1-carboximidamide hydrochloride (0.561 g, 1.60 mmol). The title compound was obtained. Yield 0.290 g (50%); m.p. 215.0–215.7 °C; IR (KBr): 3334, 3182 (N-H), 2922, 2887, 2856 (C-H), 1617, 1557, 1522 (C=N, C=C_Ar_), 1334, 1132 (SO_2_) cm^−^^1^; ^1^H NMR (500 MHz, DMSO-*d*_6_) δ: 2.27 (s, 3H, CH_3_), 3.31 (m, 4H, piperazine), 3.83 (m, 4H, piperazine and 2H, S-CH_2_), 6.74–7.53 (m, 7H, H_Ar_ and 2H, NH_2_), 7.80 (m, 1H, H-3), 7.91 (m, 1H, H-6), 10.78 (m, 1H, NH, 5-chlorobenzimidazolidine) ppm; Anal. calcd. for C_29_H_26_Cl_2_F_3_N_9_O_2_S_2_ (724.61); C, 48.07; H, 3.62; N, 17.40. Found: C, 48.00; H, 3.56; N, 17.38. HRMS (ESI-TOF) (723.0980) calcd for C_29_H_26_Cl_2_F_3_N_9_O_2_S_2_ [M + H]^+^ (724.1058) found 724.1060.

*2-[{4-Amino-6-[4-(3,4-dichlorophenyl)piperazin-1-yl]-1,3,5-triazin-2-yl}methylthio]-4-chloro-N-(5-chloro-1H-benzo[d]imidazol-2(3H)-ylidene)-5-methylbenzenesulfonamide* (**48**).

Starting from *N*-carbamimidoyl-4-(3,4-dichlorophenyl)piperazine-1-carboximidamide hydrochloride (0.445 g, 1.60 mmol). The title compound was obtained after extraction of the impurities with boiling mixture of methanol/acetonitrile (1:17.5). Yield 0.338g (58%); m.p. 205.1–205.7 °C; IR (KBr): 3329, 3218, 3180 (N-H), 2953, 2923, 2893, 2858 (C-H), 1553, 1466 (C=N, C=C_Ar_), 1286, 1130 (SO_2_) cm^−^^1^; ^1^H NMR (500 MHz, DMSO-*d*_6_) δ: 2.27 (s, 3H, CH_3_), 3.20 (m, 4H, piperazine), 3.81–3.82 (m, 4H, piperazine and 2H, S-CH_2_), 6.75–7.42 (m, 6H, H_Ar_ and 2H, NH_2_), 7.79 (m, 1H, H-3), 7.91 (m, 1H, H-6), 10.79 (m, 1H, NH, 5-chlorobenzimidazolidine) ppm; Anal. calcd. for C_28_H_25_Cl_4_N_9_O_2_S_2_ (725.50); C, 46.35; H, 3.47; N, 17.38. Found: C, 46.29; H, 3.44; N, 17.35. HRMS (ESI-TOF) (723.0327) calcd for C_28_H_25_Cl_4_N_9_O_2_S_2_ [M + H]^+^ (724.0405) found 724.0443.

*2-[{4-Amino-6-[4-(2-methoxyphenyl)piperazin-1-yl]-1,3,5-triazin-2-yl}methylthio]-4-chloro-N-(5-chloro-1H-benzo[d]imidazol-2(3H)-ylidene)-5-methylbenzenesulfonamide* (**49**).

Starting from *N*-carbamimidoyl-4-(2-methoxyphenyl)piperazine-1-carboximidamide hydrochloride (0.501 g, 1.60 mmol). The title compound was obtained after crystallization from a mixture of ethanol/acetonitrile (1:4). Yield 0.131 g (22%); m.p. 273.0–273.9 °C; IR (KBr): 3377, 3320 (N-H), 3002, 2945, 2848, 2830 (C-H), 1564, 1482 (C=N, C=C_Ar_), 1273, 1138 (SO_2_) cm^−^^1^; ^1^H NMR (500 MHz, DMSO-*d*_6_) δ: 2.32 (s, 3H, CH_3_Ph), 2.88–2.91 (m, 4H, piperazine), 3.77–3.79 (m, 4H, piperazine), 3.80 (s, 3H, O-C**H**_3_), 3.89 (s, 2H, S-CH_2_), 6.87–7.29 (m, 7H, H_Ar_ and 2H, NH_2_), 7.94 (m, 1H, H-3), 7.98 (m, 1H, H-6), 12.06 (m, 2H, NH, 5-chlorobenzimidazolidine) ppm; Anal. calcd. for C_29_H_29_Cl_2_N_9_O_3_S_2_ (686.64); C, 50.73; H, 4.26; N, 18.36. Found: C, 50.70; H, 4.21; N, 18.32. HRMS (ESI-TOF) (685.1212) calcd for C_29_H_29_Cl_2_N_9_O_3_S_2_ [M + H]^+^ (686.1290) found 686.1277.

### 3.3. Cell Culture and Cell Viability Assay

All chemicals, if not stated otherwise, were obtained from Sigma–Aldrich (St. Louis, MO, USA). The HCT-116 cell lines was purchased from ATCC (ATCC-No: CCL-247), while the MCF-7, HeLa and HaCaT cell lines were purchased from Cell Lines Services (Eppelheim, Germany). Cells were cultured in Dulbecco’s modified Eagle’s medium (DMEM) supplemented with 10% fetal bovine serum, 2 mM glutamine, 100 units/mL penicillin, and 100 µg/mL streptomycin. Cultures were maintained in a humidified atmosphere with 5% carbon dioxide at 37 °C in an incubator (Heraceus, HeraCell).

Cell viability was examined using the MTT (3-(4,5-dimethylthiazol-2-yl)-2,5-diphenyltetrazolium bromide) assay. Cells were seeded in 96-well plates at a density of 5 × 10^3^ cells/well and treated for 72 h with the tested compounds in the concentration range of 1–100 µM (1, 10, 25, 50 and 100 µM). Then, MTT (0.5 mg/mL) was added to the medium and cells were further incubated for 2 h at 37 °C. In the next stage, cells were lysed with DMSO and the absorbance of the formazan solution was measured at 550 nm with a plate reader (1420 multilabel counter, Victor, Jügesheim, Germany). The experiment was performed in triplicate. Values are expressed as the mean ± SD of at least three independent experiments. Cisplatin was used as a positive control.

### 3.4. Molecular Docking

All the molecular modeling studies were performed using Molecular Operating Environment (MOE, 2018) software. The partial charges were calculated automatically. All minimizations were performed with MOE until an RMSD gradient of 0.2534 kcal/molÅ with AMBER10 force field (a value below 2.0 kcal/molÅ indicates that the docking protocol was validated).

The X-ray crystallographic structure of MDM2 co-crystalized with Nutlin-3a (PDB ID:5C5A) was downloaded from the protein data bank available at the RCSB Protein Data Bank https://www.rcsb.org/. For each co-crystallized enzyme, water molecules and ligands that were not involved in the binding were removed. The Protonate 3D protocol in MOE with its default options was used to prepare the protein. The co-crystallized ligand (Nutlin-3a) was used to define the binding site for docking. The Triangle Matcher method was used, where 1000 poses were analyzed together with also redocking 1000 poses (to optimize docked structures) using the AMBER10 force field. From each obtained molecular docking result, five poses with the lowest energy were selected. Then one pose was selected that had the most interactions with amino acids in the MDM2 protein binding pocket. The choice of poses also took into account the number of interactions with the amino acids with which the known MDM2 (pdb: 5C5A) Nutlin-3a protein inhibitor binds. The docking scores, types of interactions and the bond lengths are shown in Table 3.

## 4. Conclusions

We have synthesized a series of novel 2-[(4-amino-6-R^2–^1,3,5-triazin-2-yl)methylthio]-4-chloro-5-methyl-*N*-(5-R^1^-1*H*-benzo[*d*]imidazol-2(3*H*)-ylidene)benzenesulfonamide **6**–**49**. The obtained compounds were tested in vitro for their cytotoxic activity, with the use of the MTT assay, toward colon (HCT-116), breast (MCF-7) and cervical (HeLa) cancer cell lines (IC_50_: 7–11µM; 15–24 µM and 11–18 µM), vs. non-cancerous cells (HaCaT) (IC_50_: 34 µM and 28 µM), respectively. The multiple linear regression technique (MLR) was applied to build up the QSAR model for predicting the cytotoxic activity of novel compounds, based on different topological (2D) and conformational (3D) molecular descriptors. Developed models showed a good predictability and might be useful for further development of structurally similar derivatives with better cytotoxic properties. The molecular docking studies revealed the possible binding mode of the most active compounds **22** and **46** within the active site of the MDM2 protein suggesting that it may be a possible molecular target for the tested compounds.

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
