# Peer review of "Synthesis, Antitumor Evaluation, Molecular Modeling and Quantitative Structure–Activity Relationship (QSAR) of Novel 2-[(4-Amino-6-N-substituted-1,3,5-triazin-2-yl)methylthio]-4-chloro-5-methyl-N-(1H-benzo[d]imidazol-2(3H)-ylidene)Benzenesulfonamides"

_ijms, 2020, doi:10.3390/ijms21082924_

Round 1
Reviewer 1 Report
The manuscript by Tomorowicz et al reports about the synthesis, cytotoxic activity and QSAR studies of a number of novel 2-[(4-amino-6-R2-1,3,5-triazin-2-yl)methylthio]-4-chloro-5-methyl-N-(5-R1-1H-benzo[d]imidazol-2(3812 H)-ylidene)benzenesulfonamide compounds derived from Nutlin-3a, a potent MDM2 inhibitor. On the basis of the crystallographic structure of the complex formed by MDM2 and Nutlin-3a the authors predict by molecular docking the putative poses of the most active compounds within the MDM2 structure. The general quality of the manuscript is good and the work fits the requirements of the journal, nevertheless, some points have to be fixed:
- The inhibition of MDM2-p53 binding should be analysed and shown to demonstrate the ability of the synthesized compounds to behave as MDM2 antagonist.
- Molecular docking: the criteria for the best pose selection is not clear. Was it the docking score or the number of interactions or the superimposition with Nutlin-3a ?
Minor points:
- Figure 5: the interactions have to be shown more clearly, the picture has to be enlarged.
Author Response
In response to the Reviewer #1
The authors are grateful for the valuable comments of the reviewers.
“The manuscript by Tomorowicz et al reports about the synthesis, cytotoxic activity and QSAR studies of a number of novel 2-[(4-amino-6-R2-1,3,5-triazin-2-yl)methylthio]-4-chloro-5-methyl-N-(5-R1-1H-benzo[d]imidazol-2(3812 H)-ylidene)benzenesulfonamide compounds derived from Nutlin-3a, a potent MDM2 inhibitor. On the basis of the crystallographic structure of the complex formed by MDM2 and Nutlin-3a the authors predict by molecular docking the putative poses of the most active compounds within the MDM2 structure. The general quality of the manuscript is good and the work fits the requirements of the journal, nevertheless, some points have to be fixed:”
“The inhibition of MDM2-p53 binding should be analysed and shown to demonstrate the ability of the synthesized compounds to behave as MDM2 antagonist.”
Reports from the scientific literature clearly indicate that 1,2,4,5-tetrasubstituted 4,5-cis-imidazolines are recognized MDM2 protein inhibitors [12,13] and also Annual Reports in Medicinal Chemistry, vol. 49 (2014) 167-187. The synthesized compounds are their structural analogues, hence in the light of the conducted research it seems that the probable mechanism of their anti-tumor activity is the inhibition of MDM2 protein. Moreover, we carried out molecular docking for various (over then 25) other molecular targets associated with tumors that not showed the affinity of designed compounds equal or better than for the MDM2 protein.
“Molecular docking: the criteria for the best pose selection is not clear. Was it the docking score or the number of interactions or the superimposition with Nutlin-3a ?”
We explain that docking was performed using the Triangle Matcher method where 1000 poses were analyzed together with redocking also 1000 poses (to optimize docked structures) using the AMBER10 force field. From each obtained molecular docking result, 5 poses with the lowest energy were selected. Then one was selected that had the most interactions with amino acids in the MDM2 protein binding pocket. The choice of poses also took into account the number of interactions with the amino acids with which the known MDM2 (pdb: 5C5A) Nutlin-3a protein inhibitor binds - the number of found interactions is seven.
“Minor points:
Figure 5: the interactions have to be shown more clearly, the picture has to be enlarged.”
Figure 5 was significantly enlarged, showing clearly protein-ligand interactions inside the active site of MDM2.
Reviewer 2 Report
Dear authors,
I've read the manuscript and analyzed the details of its presentation. The manuscript is overall well written and will certainly be of interest for the part of the Journal readership. Here are a few comments, which can help you to improve the quality of the manuscript in my view.
- Introduction. As I can see, the synthesis of the new chemicals is based on the structures, reported earlier (Molecules, 2015). Therefore it would be useful if the authors emphasize the novelty of the synthesized molecules both from chemical and biological approaches (the novelty of chemical synthesis and the novelty of biological mechanism).
- Methods. It is not completely clear why the mechanism of antitumor activity for the reported novel substances is the inhibition of mdm2 protein. Multiple mechanisms of antineoplastic activity may exist. The authors should either carefully explain the rationale of this hypothesis or provide the evidence for them from the literature. I would like to add that the inhibition of mdm2 or disruption of the p53-mdm2 complex in only one of the possible mechanisms of any antineoplastic activity of the drug. And since the authors do not provide experiments that can prove the inhibition of the mdm2 by the reported compounds, it is difficult to associate their activity with the inhibition of mdm2.
- The authors should explain why they chose linear QSAR equations which demonstrate rather average results instead of using nonlinear models which can help to obtain higher accuracy. In fact, it is not clear what is the purpose of QSAR in the paper. The paragraph, describing QSAR contains the enumeration of the descriptors used for building the model without any analysis of its impact on the general idea of the study.
With best regards.
Author Response
In response to the Reviewer #2
The authors are grateful for the valuable comments of the reviewers.
“I've read the manuscript and analyzed the details of its presentation. The manuscript is overall well written and will certainly be of interest for the part of the Journal readership. Here are a few comments, which can help you to improve the quality of the manuscript in my view.
- Introduction. As I can see, the synthesis of the new chemicals is based on the structures, reported earlier (Molecules, 2015). Therefore it would be useful if the authors emphasize the novelty of the synthesized molecules both from chemical and biological approaches (the novelty of chemical synthesis and the novelty of biological mechanism).”
We are grateful for valuable comments. According to these comments, we have improved our introduction with information emphasizing a novelty aspect of described compounds. The following fragment has been added to the manuscript: “Our previous works on search for antitumor agents among benzenesulfonamide derivatives carried out by Sławinski's group indicate the importance of 2-methylthiobenzenesulfonamide fragment for cytotoxic activity of compounds against cervical, breast and colon cancer. We have proved that our compounds showed the apoptotic effect in cancer cells (Molecules 2016, 21. 808, Chem. Biol. & Drug Des. 2017, 90, 380-396, Eur. J. Med. Chem. 2018, 155, 670-680, Continuing the search for more active compounds, we designed and developed a method for the synthesis of new molecules with potential inhibitory activity against MDM2 protein. We carried out molecular docking for various targets associated with tumors that showed the affinity of designed compounds for MDM2.”
- “Methods. It is not completely clear why the mechanism of antitumor activity for the reported novel substances is the inhibition of mdm2 protein. Multiple mechanisms of antineoplastic activity may exist. The authors should either carefully explain the rationale of this hypothesis or provide the evidence for them from the literature. I would like to add that the inhibition of mdm2 or disruption of the p53-mdm2 complex in only one of the possible mechanisms of any antineoplastic activity of the drug. And since the authors do not provide experiments that can prove the inhibition of the mdm2 by the reported compounds, it is difficult to associate their activity with the inhibition of mdm2.”
We agree with Reviewer and according to Reviewer’s comment we added fragment explaining rationale of our hypothesis. “We carried out molecular docking for various targets associated with tumors that showed the affinity of designed compounds for MDM2.”
Reports from the scientific literature clearly indicate that 1,2,4,5-tetrasubstituted 4,5-cis-imidazolines are recognized MDM2 protein inhibitors [12,13] and also Annual Reports in Medicinal Chemistry, vol. 49 (2014) 167-187. The synthesized compounds are their structural analogues, hence in the light of the conducted research it seems that the probable mechanism of their anti-tumor activity is the inhibition of MDM2 protein. Moreover, we carried out molecular docking for various (over then 25) other molecular targets associated with tumors that not showed the affinity of designed compounds equal or better than for the MDM2 protein.
- “The authors should explain why they chose linear QSAR equations which demonstrate rather average results instead of using nonlinear models which can help to obtain higher accuracy. In fact, it is not clear what is the purpose of QSAR in the paper. The paragraph, describing QSAR contains the enumeration of the descriptors used for building the model without any analysis of its impact on the general idea of the study”
The use of linear models was justified by the fact that the distribution of activity in the range of interest to us up to a maximum of 50 µM shows approximately linear nature, what was checked at the beginning by us.
Linear regression is a method that copes well with interpolation, and our goal was to generate models that predict the activity of new compounds in the range up to about 50 µM. Accurate prediction of higher values over 50 µM is not the goal of the generated models.
The main purpose of QSAR analysis was to generate models in which it would be possible to predict in silico the activity of compounds planned for synthesis, in other words computer aided design of drugs. As we can see in Fig. 3, the generated models well predicted activity of test set – red points for all tested cell lines.
In addition, the next purpose of generating the models was to determine which of easily identifiable molecular characteristics influence on the antitumor activity of the synthesized compounds. We put such information in the manuscript on pages 6-7 in lines 165 to 177. If the model is built using linear methods, it can be easily stated that the activity increases or decreases with the increase or decrease of the value of each descriptor. For example, it can be seen from the generalized linear equation that the activity for the HCT-116 line increases if we reduce the number of oxygen atoms or for the HeLa line the activity increases if we increase the number of fluorine atoms.
We hope that the manuscript in the present shape will be suitable for publication in International Journal of Molecular Science.
With best regards,
Prof. Jarosław Sławiński
Round 2
Reviewer 1 Report
The authors try to explain, in the letter, the docking protocol which in the manuscript is limited to the employed method (Triangle Matched) and a vague "The interactions of ligand with the amino acids of the active site of MDM2 enzyme were studied. Only one pose was selected for each compound and the selection was based on the number of interactions with the enzyme, docking score". However no changes have been introduced in the manuscript wherease it is important to explain to the reader, step by step, how the best pose was selected.
Author Response
In response to the Reviewer #1
The authors are grateful for the valuable comments of the reviewers.
The authors try to explain, in the letter, the docking protocol which in the manuscript is limited to the employed method (Triangle Matched) and a vague "The interactions of ligand with the amino acids of the active site of MDM2 enzyme were studied. Only one pose was selected for each compound and the selection was based on the number of interactions with the enzyme, docking score". However no changes have been introduced in the manuscript wherease it is important to explain to the reader, step by step, how the best pose was selected.
We introduced in paragraph 3.3. “Molecular docking” on page 23 the following fragment:
“The Triangle Matcher method was used, where 1000 poses were analyzed together with redocking also 1000 poses (to optimize docked structures) using the AMBER10 force field. From each obtained molecular docking result, 5 poses with the lowest energy were selected. Then one pose was selected that had the most interactions with amino acids in the MDM2 protein binding pocket. The choice of poses also took into account the number of interactions with the amino acids with which the known MDM2 (pdb: 5C5A) Nutlin-3a protein inhibitor binds. The docking scores, types of interactions and the bond lengths are shown in Table 4.
Reviewer 2 Report
Dear authors,
thank you for your comments, the corrections are acceptable.
I still think that the authors can use more complex QSAR equations in their future studies.
With best regards.
Author Response
In response to the Reviewer #2
Dear authors, thank you for your comments, the corrections are acceptable.
I still think that the authors can use more complex QSAR equations in their future studies.
The authors are very grateful for the valuable comments of the reviewer and for positive opinion.